# Daam2 driven degradation of VHL promotes gliomagenesis

**Wenyi Zhu[1,2], Saritha Krishna[1†], Cristina Garcia[1†], Chia-Ching John Lin[1], Bartley D Mitchell[3], Kenneth L Scott[4], Carrie A Mohila[5], Chad J Creighton[6,7], Seung-Hee Yoo[8], Hyun Kyoung Lee[9,10,11]\*, Benjamin Deneen[1,2,10,11]\***

[1]Center for Cell and Gene Therapy, Baylor College of Medicine, Houston, United States; [2]The Integrative Molecular and Biomedical Sciences Graduate Program, Baylor College of Medicine, Houston, United States; [3]Department of Neurosurgery, Baylor College of Medicine, Houston, United States; [4]Department of Human and Molecular Genetics, Baylor College of Medicine, Houston, United States; [5]Department of Pathology, Texas Children's Hospital, Houston, United States; [6]Dan L Duncan Cancer Center, Division of Biostatistics, Baylor College of Medicine, Houston, United States; [7]Department of Medicine, Baylor College of Medicine, Houston, United States; [8]Department of Biochemistry and Molecular Biology, The University of Texas Heath Science Center at Houston, Houston, United States; [9]Department of Pediatrics, Division of Neurology, Baylor College of Medicine, Houston, United States; [10]Neurological Research Institute, Texas Children's Hospital, Houston, United States; [11]Department of Neuroscience, Baylor College of Medicine, Houston, United States

**\*For correspondence:**
hyunkyol@bcm.edu (HKL);
deneen@bcm.edu (BD)

[†]These authors contributed equally to this work

**Competing interests:** The authors declare that no competing interests exist.

**Abstract** Von Hippel-Landau (VHL) protein is a potent tumor suppressor regulating numerous pathways that drive cancer, but mutations in VHL are restricted to limited subsets of malignancies. Here we identified a novel mechanism for VHL suppression in tumors that do not have inactivating mutations. Using developmental processes to uncover new pathways contributing to tumorigenesis, we found that Daam2 promotes glioma formation. Protein expression screening identified an inverse correlation between Daam2 and VHL expression across a host of cancers, including glioma. These in silico insights guided corroborating functional studies, which revealed that Daam2 promotes tumorigenesis by suppressing VHL expression. Furthermore, biochemical analyses demonstrate that Daam2 associates with VHL and facilitates its ubiquitination and degradation. Together, these studies are the first to define an upstream mechanism regulating VHL suppression in cancer and describe the role of Daam2 in tumorigenesis.

DOI: https://doi.org/10.7554/eLife.31926.001

## Introduction

Tumor suppressor and oncogenic pathways function in part by subverting existing cellular programs to promote cancer 'hallmark' properties that engender malignant growth (*Hanahan and Weinberg, 2011*). This corruption of normal cellular physiology is mediated by aberrant activities of these tumorigenic pathways, which are predominantly driven by genetic mutation. Importantly, genetic mutation is not the sole source of this dysregulation, as changes in gene expression via promoter methylation or protein turnover can phenotypically resemble driver mutations and contribute to tumorigenesis (*Esteller et al., 1999*; *Hegi et al., 2004*; *Pineda et al., 2015*; *Reinstein and Ciechanover, 2006*; *Semenza, 2003*; *Shen et al., 2005*; *Zöchbauer-Müller et al., 2001*). While these broad

**eLife digest** Glioblastoma is the deadliest form of brain cancer, and the rate of patient survival has not significantly improved over the past 70 years. This cancer arises when glial cells, which provide support and insulation to nerve cells, develop mutations that alter the activity of certain genes or alter the role they play in cells. However, there are also several key genes linked to glioblastomas that don't exhibit mutations, such as the gene that encodes the Von Hippel Landau protein (or VHL for short). This protein normally helps to protect us from developing cancer, but it is not clear how this protein is prevented from performing this role in glioblastomas.

One possibility is that proteins that regulate how cells grow and develop may control VHL. For example, a protein called Daam2 plays a critical role in a signaling pathway that is required for glial cell development. Zhu et al. used biochemical techniques to study Daam2 and VHL in both human cells and mouse models of glioblastoma.

The experiments show that glioblastoma cells have lower levels of VHL compared to normal cells. This decrease is caused by Daam 2, which interacts with VHL and promotes its degradation. Further experiments found that in several different types of cancer, higher levels of Daam2 are linked with the presence of lower levels of VHL.

These findings indicate that the interaction between Daam2 and VHL could be a new target for drugs to treat glioblastoma and possibly other forms of cancer. Daam2 and VHL have also been linked to multiple sclerosis, cerebral palsy and other diseases that affect the nervous system. Therefore, understanding how these proteins interact may also help to develop new treatments for these conditions.

DOI: https://doi.org/10.7554/eLife.31926.002

regulatory processes have been linked to cancer, the underlying molecular mechanisms that regulate expression of key components of tumorigenic pathways are very poorly characterized.

*VHL* is a key tumor suppressor that is mutated in Von Hippel-Landau disease, a hereditary cancer predisposition syndrome that often manifests as clear-cell renal carcinoma (ccRCC) (*Chen et al., 1995*; *Gossage et al., 2015*; *Kim and Kaelin, 2004*; *Maher et al., 1990*). VHL functions by binding to HIF1$\alpha$ and hydroxylated Akt and modulating their degradation and activity, respectively. Mutant forms of *VHL* associated with ccRCC are incapable of binding HIF1$\alpha$ or pAkt, resulting in stabilized expression and activation of these proteins, which ultimately facilitates tumorigenesis (*Guo et al., 2016*; *Ivan et al., 2001*; *Jaakkola et al., 2001*; *Maxwell et al., 1999*; *Min et al., 2002*; *Ohh et al., 2000*). While dysregulated HIF1$\alpha$ and pAkt are associated with most forms of cancer, mutations in VHL are found predominately in ccRCC. This dichotomy suggests that additional regulatory mechanisms oversee VHL dysregulation or inactivation in other malignancies. Indeed recent studies in glioma have shown that ID2 can interfere with VHL activity in cell lines (*Lee et al., 2016*) and that it's subject to regulation by microRNAs (*Li et al., 2017*). Nevertheless, the upstream mechanisms that directly regulate *VHL* expression and protein turnover in cancer remain poorly defined.

One potential mode of tumor suppressor gene regulation is through developmental mechanisms. Developmental processes directly contribute to all forms of malignancy and are utilized by tumorigenic pathways to maintain cells in an undifferentiated and proliferative state (*Jackson et al., 2006*; *Kesari et al., 2005*; *Stiles and Rowitch, 2008*). Given these established molecular and functional interactions, it stands to reason that expression of tumorigenic pathways may be reciprocally regulated by developmental mechanisms. However, whether such reciprocal regulation of tumor suppressor pathways by developmental factors contributes to tumorigenesis is poorly defined.

To investigate the interface between developmental programs and the regulation of tumor suppressor pathways, we used malignant glioma as a model system. As a molecular entry point for these studies, we focused on Daam2, a key developmental regulator that suppresses glial differentiation and also contributes to dorsal patterning in the developing CNS (*Lee et al., 2015*; *Lee and Deneen, 2012*). Here we found that Daam2 promotes tumorigenesis in mouse and human models of malignant glioma. Bioinformatics analysis revealed that Daam2 and VHL expression is inversely correlated across a host of human malignancies. These in silico observations are corroborated by in vivo functional studies, which revealed that Daam2 promotes tumorigenesis by suppressing VHL expression.

Mechanistically, we found that Daam2 associates with VHL and facilitates its degradation by the ubiquitin pathway. Together, these studies represent the initial characterization of Daam2 function in glioma and define, for the first time, an upstream regulatory mechanism that controls VHL protein expression in cancer. Moreover, because mutations in VHL are restricted to a limited set of malignancies, we have identified a new mechanism for VHL inactivation in tumors that do not have inactivating mutations.

## Results

### Daam2 is expressed in human and mouse glioma

Previously, we identified Daam2 as a component of the Wnt receptor complex that directly contributes to dorsal patterning in the embryonic spinal cord and oligodendrocyte differentiation during development and after injury (*Lee et al., 2015*; *Lee and Deneen, 2012*). To further explore its role in neurological diseases, we sought to investigate its role in malignancies of the CNS. Towards this we took advantage of the TCGA pan-cancer expression data set (*Akbani et al., 2014*; *Weinstein et al., 2013*) and evaluated Daam2 expression across a spectrum of 34 malignancies, finding that it's most highly expressed in low-grade glioma (LGG) and glioblastoma multiforme (GBM) (*Figure 1A*).

To validate DAAM2 expression in human LGG and GBM, we used in situ hybridization (ISH) across a cohort of 35 LGG and 40 GBM primary human samples, finding that DAAM2 demonstrates heterogeneous expression within each glioma sub-type (*Figure 1B–C*; *Figure 1—figure supplement 1*). Notably, the majority of tumors exhibited staining intensity scores that exceeded normal brain samples, indicating that Daam2 expression is elevated within glioma tumors. In addition, we assessed DAAM2 expression via qRT-PCR in primary human glioma samples and non-tumor white matter, and consistent with the tissue microarray data, we found that Daam2 expression is elevated in a majority of these primary samples (*Figure 1—figure supplement 1*). Next, we evaluated Daam2 expression in our mouse model of malignant glioma, where we combine in utero electroporation (IUE), with CRISPR-mediated deletion of *Nf1*, *Pten*, and *Trp53* (herein CRISPR/IUE) (*Figure 2—figure supplement 1*) (*Chen et al., 2016*; *Chen and LoTurco, 2012*; *Chen et al., 2012*; *Cong et al., 2013*; *John Lin et al., 2017*). This model closely resembles the genetics of GBM (*Alcantara Llaguno et al., 2009*; *Kwon et al., 2008*; *Zhu et al., 2005*) and begins producing tumors detectable around postnatal week 8. Combining this model, with our the Daam2-LacZ mouse line, we performed immunostaining on the resultant tumors, finding that Daam2 exhibits elevated expression levels in tumors, compared to normal brain tissue (*Figure 1D*; *Figure 2—figure supplement 1*). Finally, we evaluated Daam2 expression in xenograft tumors generated from primary human GBM cell lines, finding that Daam2 is also highly expressed in these human cell line models (*Figure 1—figure supplement 1*). Put together, these studies indicate that Daam2 expression is elevated in both human LGG and GBM and is expressed in the associated model systems.

### Overexpression of Daam2 accelerates glioma tumorigenesis

The expression analysis in human glioma and associated mouse models led us to examine whether Daam2 contributes to glioma tumorigenesis. To assess its role in glioma, we performed overexpression, gain-of-function (GOF) studies in human GBM cell lines, finding that Daam2 accelerates the rate of cell growth in vitro (*Figure 2A–B,F*). Next, we determined how Daam2 influences anchorage independent growth, via soft agar assay, finding that it also accelerates colony formation (*Figure 2C–E*). Together, these in vitro studies, indicate that Daam2 promotes cell proliferation and growth in human GBM cell lines

To evaluate the role of Daam2 in tumorigenesis, we turned to mouse models of malignant glioma. The first model combines targeted PiggyBac overexpression of oncogenic Ras-V12 (PB-Ras) in glial precursors with IUE, to generate malignant glioma in mice around post-natal day 14 (*Figure 2—figure supplement 1*). Combining PiggyBac-mediated overexpression of Daam2 with this Ras-driven model resulted in an acceleration of tumorigenesis (*Figure 2—figure supplement 2*). To further substantiate these findings in additional mouse models, we next used our CRISPR/IUE model (*Figure 2—figure supplement 1*). Consistent with the Ras model, combined overexpression of Daam2 in the CRISPR/IUE model, also resulted in accelerated tumorigenesis (*Figure 2G–I*). BrdU labeling analysis

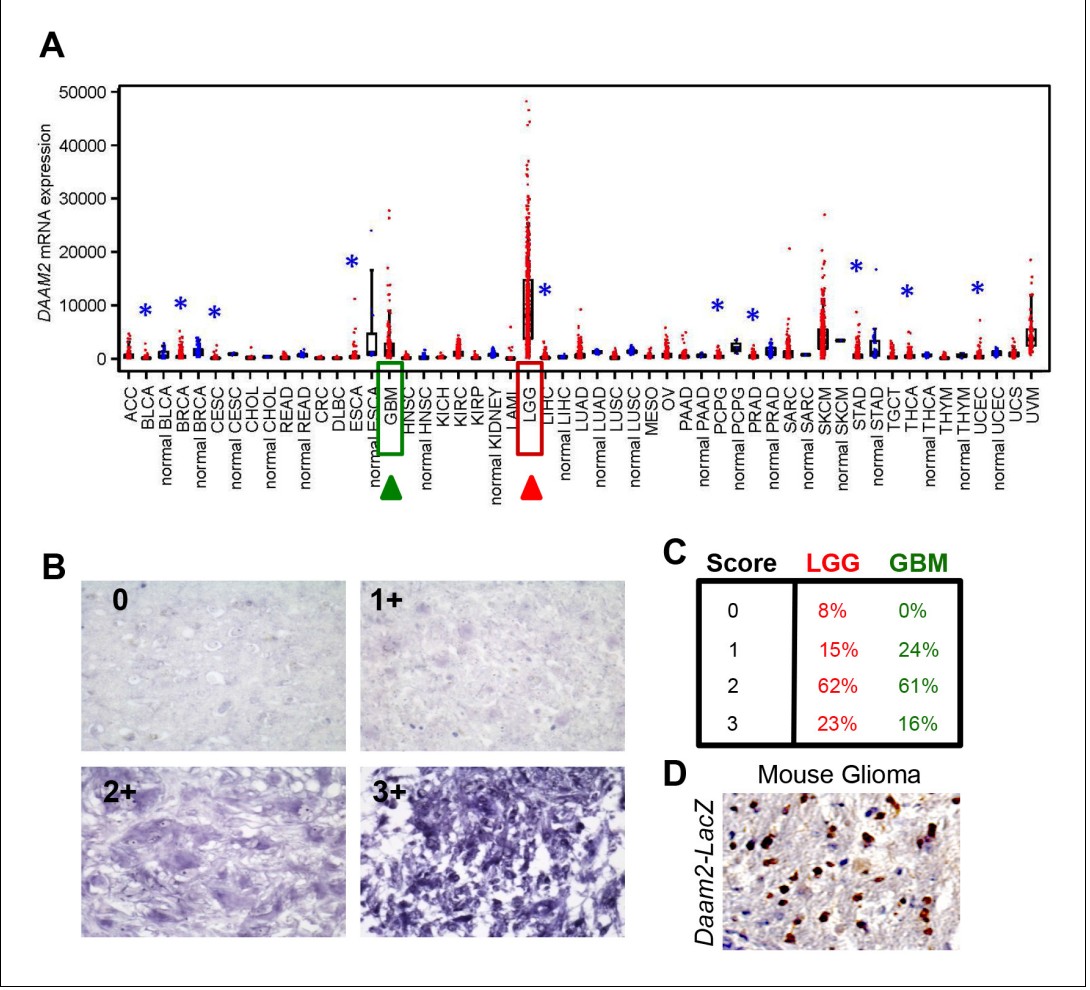

**Figure 1.** Daam2 is expressed in human glioma. (**A**) Analysis of Daam2 RNA expression across a spectrum of cancers. Data was generated by TCGA Research Network and is publicly available (see methods). Box plots represent 5%, 25%, 50%, 75%, and 95%. Asterisk denotes tumor groups significantly higher (red) or lower (blue) as compared to corresponding non-tumor ('normal') group. P-values were generated by t-test on log-transformed data. (**B**) Examples of pathological grading of in situ hybridization analysis of Daam2 expression in human glioma tissue microarrays. Note that a score of 0 demonstrates no expression, while a score of 3 denotes very high expression. (**C**) Graded pathological scoring (0-low intensity, 3-high intensity) of Daam2 expression in the samples in B; values are the percentage of LGG and GBM tumors from the tissue microarray that were assigned respective score. (**D**) Immunohistochemistry analysis of Daam2-LacZ expression in mouse glioma generated in Daam2^LacZ/+ mice.

DOI: https://doi.org/10.7554/eLife.31926.003

The following figure supplement is available for figure 1:

**Figure supplement 1.** Daam2 is expressed in experimental models of human glioma.
DOI: https://doi.org/10.7554/eLife.31926.004

of the resulting tumors from both models revealed that overexpression of Daam2 results in an increase in the number BrdU-expressing cells (*Figure 2J–L*; *Figure 2—figure supplement 2*). These data, combined with our in vitro studies, indicate that overexpression of Daam2 promotes glioma cell proliferation and tumorigenesis.

## Loss of Daam2 impairs glioma tumorigenesis

To further delineate the role of Daam2 in glioma, we next performed a series of complementary loss-of-function (LOF) studies in human and mouse glioma models. In human GBM cell lines we performed shRNAi-mediated knockdown of human Daam2, finding that decreased expression of

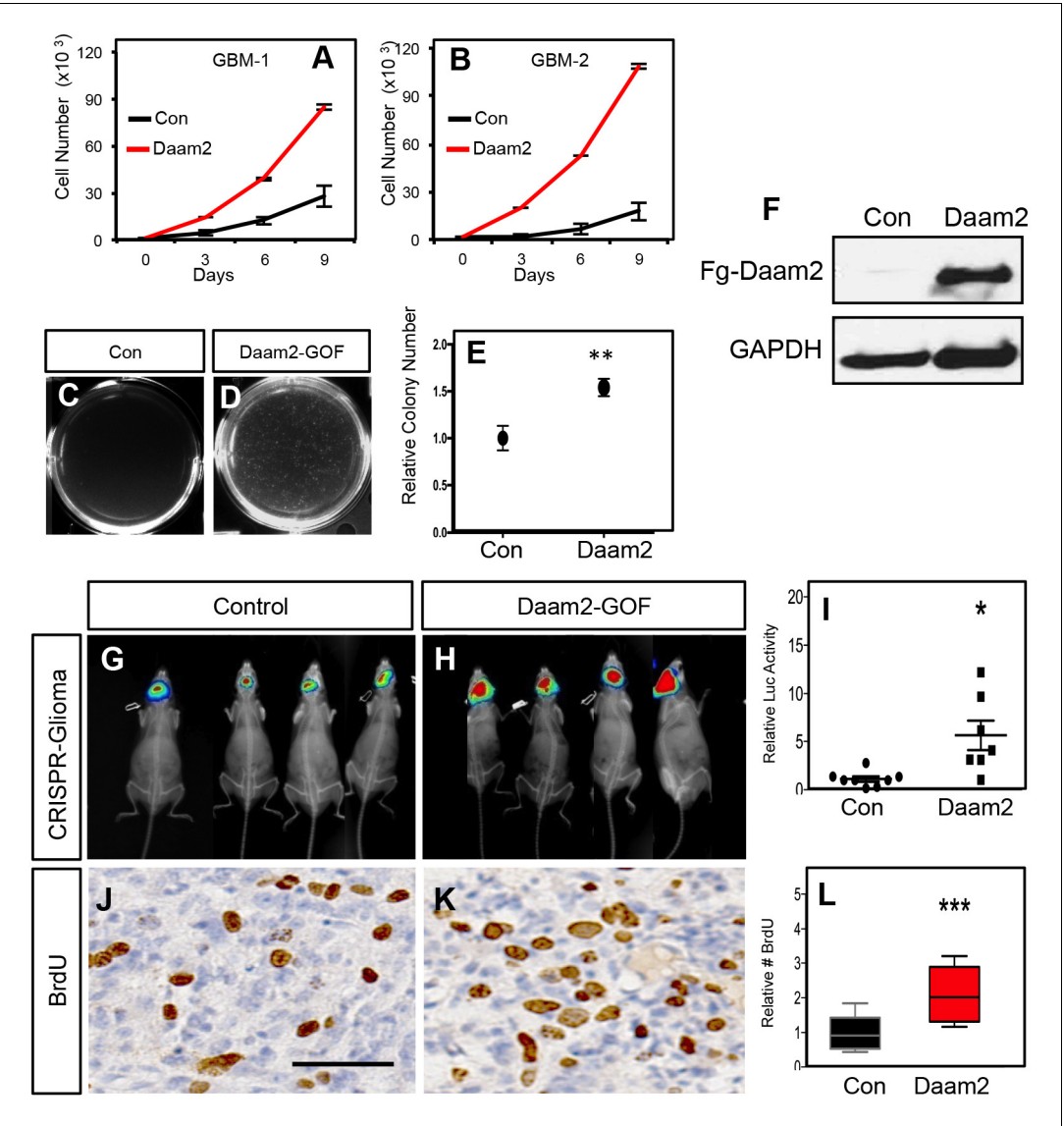

**Figure 2.** Daam2 accelerates glioma tumorigenesis. (**A–B**) Cell proliferation analysis of human GBM1 and GBM2 cell lines infected with lentivirus containing Daam2 or GFP control. (**C–D**) Soft agar assay with GBM1 cell line infected with lentivirus containing Daam2 or GFP control, images are representative. Each of these in vitro experiments was performed in triplicate and replicated three times; quantification is in E, **p=0.036. (**F**) Representative immuoblots showing ectopic expression of flag-tagged Daam2 in GBM-1 cell line. (**G–I**) Representative bioluminescence imaging of mice bearing CRISPR-IUE glioma with Daam2 overexpression or control, imaged at 7 weeks of age. Imaging quantification in I is derived from 7 Daam2-overexpression and eight control mice; *p=0.0079. (**J–L**) Immunohistochemistry analysis of BrdU expression from mice bearing CRISPR-IUE glioma from each experimental condition. Relative number of BrdU expressing cells is quantified in L and is derived from six total mice, four slides per mouse, from each experimental condition; ***p=0.0019. Scale bar in J is 50 nm. Error bars in E, I and L are ± SEM.

DOI: https://doi.org/10.7554/eLife.31926.005

The following figure supplements are available for figure 2:

**Figure supplement 1.** Mouse glioma model systems.
DOI: https://doi.org/10.7554/eLife.31926.006

**Figure supplement 2.** Daam2 accelerates glioma tumorigenesis in PB-Ras model.
DOI: https://doi.org/10.7554/eLife.31926.007

Daam2 inhibited their rate of growth in vitro (*Figure 3A–B*; *Figure 3—figure supplement 1*). Next, we assessed the tumorigenic potential of these GBM cell lines, finding that knockdown of Daam2 resulted in a significant decrease in tumor growth in vivo (*Figure 3C–E*). These knockdown studies across both in vitro and in vivo systems, complement the overexpression studies, and further substantiate the role of Daam2 in glioma cell proliferation and tumorigenesis.

To further evaluate the necessity for Daam2 in glioma tumorigenesis, we used CRISPR/IUE model to generate malignant glioma in *Daam2$^{+/-}$* and *Daam2$^{-/-}$* mice (*Lee et al., 2015*). As shown in *Figure 3F–H*, mice lacking Daam2 demonstrated decreased rates of tumor formation in this model compared to the heterozygote control. Moreover, BrdU labeling revealed substantial decreases in the number of proliferating cells in *Daam2$^{-/-}$* tumors (*Figure 3I–K*). Together, our LOF, and complementing GOF (*Figure 2*) studies in human and mouse models of glioma indicate that Daam2 promotes cell proliferation and tumorigenesis.

## Daam2 suppresses VHL expression

Having established that Daam2 functions to promote glioma tumorigenesis, we next sought to uncover the mechanism by which it operates. Previously, we found that Daam2 functions as a positive regulator of Wnt-signaling in the developing CNS (*Lee and Deneen, 2012*), suggesting that it may also function in this manner in glioma. To evaluate Wnt-activity we used an established Wnt-reporter (TOP-FLASH) and found that modulation of Daam2 expression has a modest effect on Wnt activity in glioma cell lines and did not impact Wnt activity in our mouse model of glioma (*Figure 4—figure supplement 1*).

That Wnt-activity is not affected by changes in Daam2 expression raises the question of how Daam2 promotes glioma tumorigenesis. To identify changes in protein expression associated with Daam2-mediated tumorigenesis we performed reverse phase protein lysate microarray (RPPA) on FACS-isolated mouse glioma samples that overexpress Daam2. Analysis revealed a cohort of proteins that are strongly downregulated in glioma samples that overexpress Daam2 (*Figure 4A*; *Figure 4—figure supplement 1*; *Supplementary file 1*). To further substantiate these potential relationships, we leveraged existing TCGA RNA-Seq and RPPA data to determine whether there is an inverse correlation between Daam2 expression and this cohort of proteins across a spectrum of human malignancies (*Figure 4—figure supplement 2*). This analysis identified von Hippel Landau (VHL) as one of the proteins from this cohort with the most significant inverse correlation score across this spectrum of malignancies, with lung adenocarcinoma and GBM demonstrating the strongest negative correlations between Daam2 and VHL (*Figure 4B*; *Figure 4—figure supplement 2*).

VHL is an established tumor suppressor that is frequently mutated in clear cell renal carcinoma (ccRCC) and functions by facilitating the degradation of HIF1$\alpha$ and pAkt (*Guo et al., 2016*; *Ivan et al., 2001*; *Jaakkola et al., 2001*; *Maxwell et al., 1999*; *Min et al., 2002*; *Ohh et al., 2000*). Given that Daam2 expression is inversely correlated with VHL, we next assessed whether Daam2 demonstrates congruent correlation with HIF1$\alpha$ and pAkt in the panel of human cancers. Analysis of these data revealed that Daam2 expression was positively correlated with HIF1$\alpha$ and pAkt protein expression across this cohort of malignancies, with GBM and lung adenocarcinoma also showing the strongest positive correlation (*Figure 4B*). Together, our RPPA screen and associated bioinformatics analysis of human malignancies indicate that Daam2 expression is inversely correlated with VHL expression and positively correlated with its downstream signaling axis.

## Daam2 promotes tumorigenesis through VHL

The RPPA data suggests that Daam2 modulates expression of VHL in glioma. To directly test this hypothesis, we evaluated expression of VHL and its signaling axis in our Daam2-GOF and Daam2-LOF mouse tumor models. Immunostaining for VHL in these models corroborated our RPPA data, where VHL expression was dramatically reduced in Daam2-GOF tumors, and increased in the *Daam2$^{-/-}$* tumors (*Figure 4C–F*). In addition, we also observed elevated levels of VHL protein in the brains of Daam2-/- mice, further reinforcing these expression dynamics (*Figure 4—figure supplement 2*). Next we assessed expression of VHL's downstream effectors, pAkt and HIF1$\alpha$, finding that both of these proteins demonstrated elevated levels of expression in the Daam2-GOF tumors, and pAkt having decreased levels in the *Daam2$^{-/-}$* tumors (*Figure 4G–K*). One prediction of elevated HIF1$\alpha$ is a concomitant increase in the expression of its target genes. Therefore we assessed the

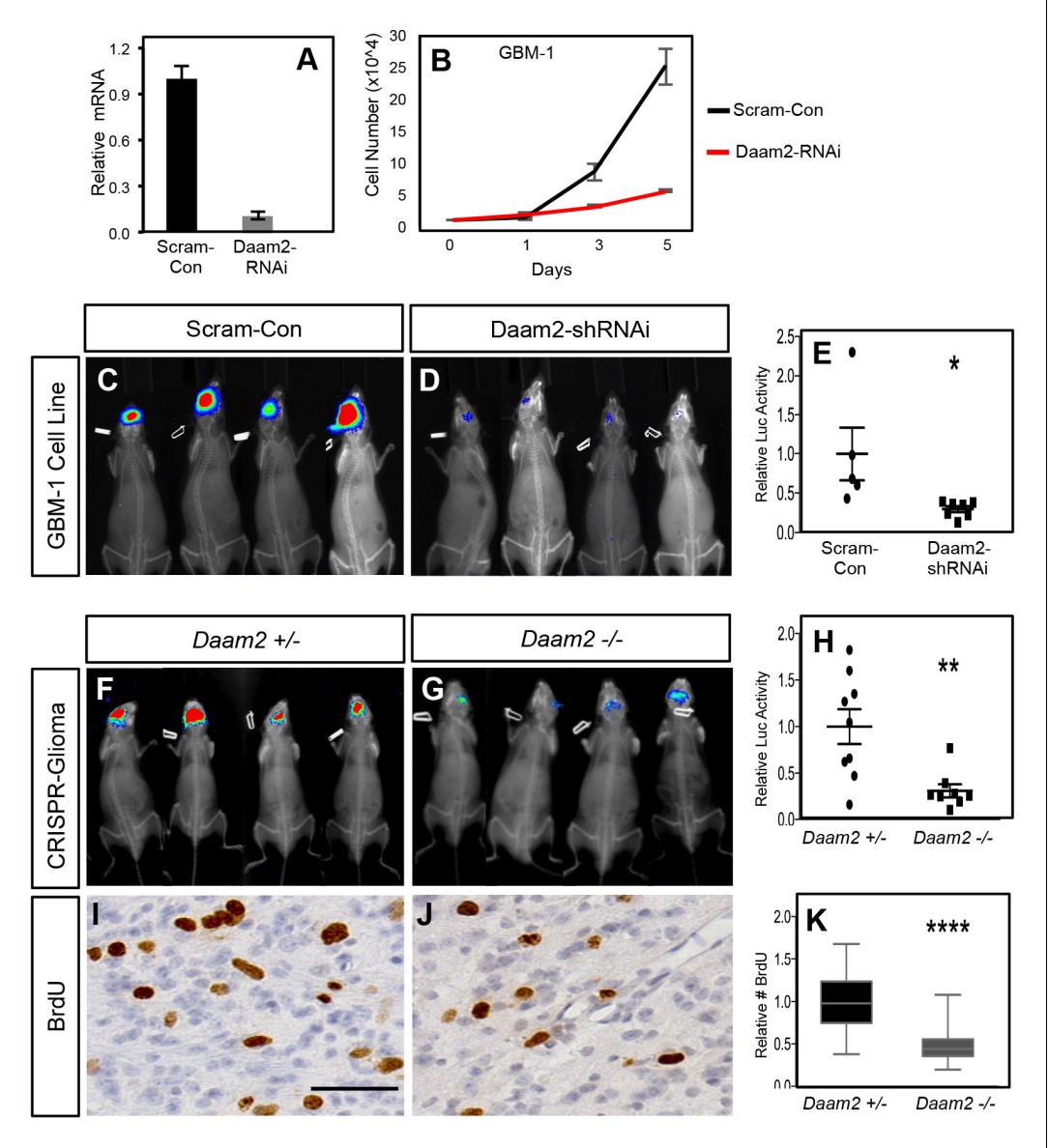

**Figure 3.** Loss of Daam2 suppresses glioma tumorigenesis. (A) qRT-PCR demonstrating effective knockdown of human Daam2 mRNA expression in human GBM1 cell line infected with lentivirus containing Daam2-shRNAi or scrambled control. (B) Cell proliferation analysis of human GBM1 cell line infected with human Daam2-shRNAi or scrambled control lentivirus. (C–E) Representative bioluminescence imaging of mice transplanted with GBM1 cell lines transfected with Daam2-shRNAi or scrambled control, imaged 6 weeks post-transplantation. Imaging quantification is derived from seven mice transplanted with GBM1/Daam2-shRNAi and 5 GBM1/scrambled controls; *p=0.0325. (F–H) Representative bioluminescence imaging of mice bearing CRISPR-IUE glioma generated in *Daam2* ± or *Daam2-/-* mice, imaged at 8 weeks of age. Imaging quantification is derived from 8 *Daam2-/-* and 9 *Daam2* ± mice; **p=0.0049. (I–K) Immunohistochemistry analysis of BrdU expression from mice bearing CRISPR-IUE glioma from each experimental condition. Relative number of BrdU expressing cells is quantified in K and is derived from six total mice, four slides per mouse, from each experimental condition; ****p=0.00001. Scale bar in I is 50 nm. Error bars in E, H and K are ± SEM.

DOI: https://doi.org/10.7554/eLife.31926.008

The following figure supplement is available for figure 3:

**Figure supplement 1.** Rescue of Daam2-shRNAi growth defects in GBM cell lines.
DOI: https://doi.org/10.7554/eLife.31926.009

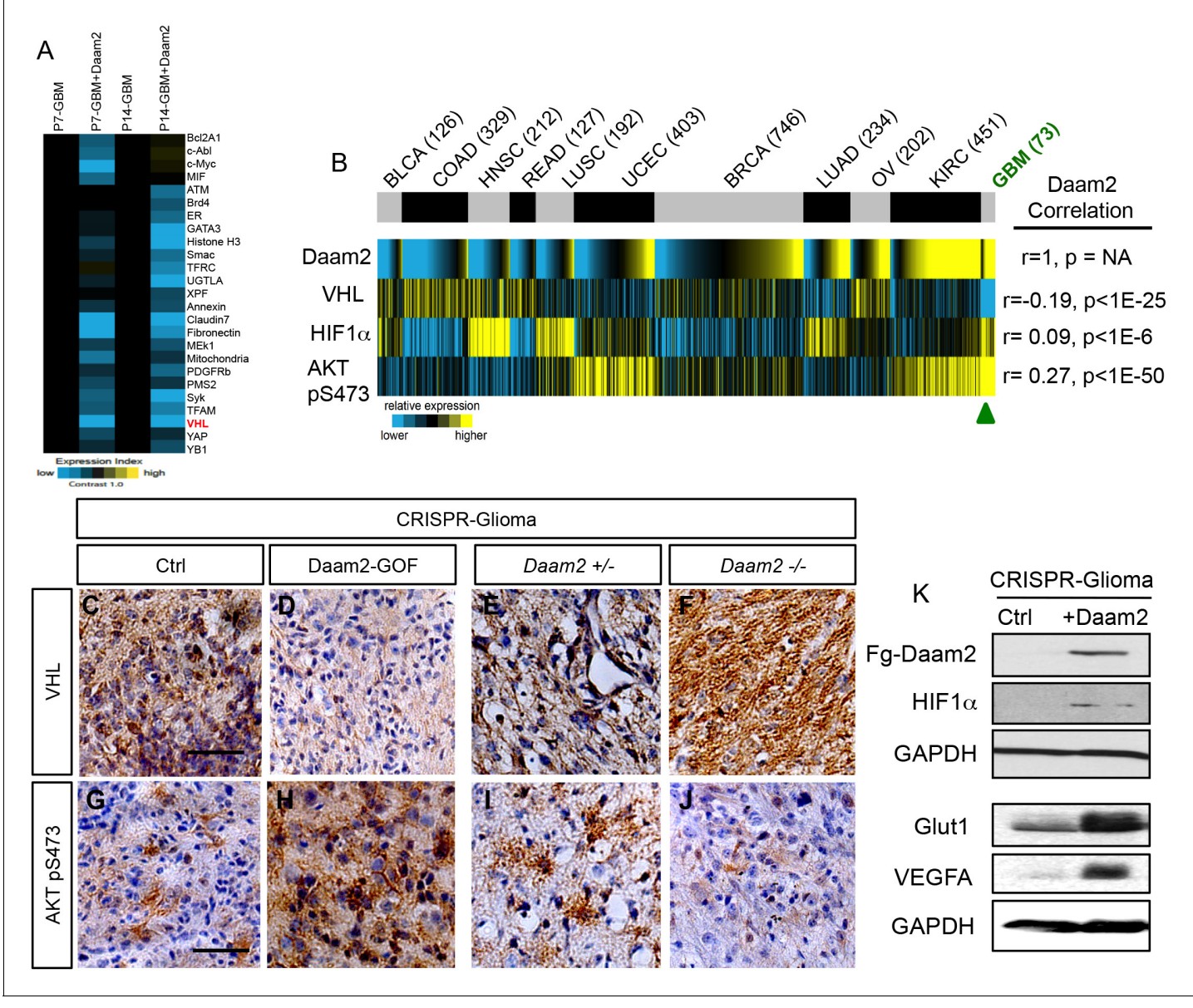

**Figure 4.** Daam2 suppresses VHL expression. (**A**) Heatmap analysis of RPPA data showing the core cohort of proteins downregulated by overexpression of Daam2 in the PB-Ras mouse glioma model. (**B**) Correlation of Daam2 mRNA expression with VHL and Akt pS473 protein expression data and HIF1α mRNA expression. Protein and mRNA expression data from samples was obtained from The Cancer Proteome Atlas (TCPA) and correlations with Daam2 mRNA was performed using Pearson's. Abbreviated cancer type is listed on top panel and number of tumors analyzed in the TCPA dataset for each cancer type is listed in parenthesis; GBM is denoted by green text. Cancer type abbreviations: BLCA: bladder urothelial carcinoma, COAD: Colon Adenocarcinoma, HNSC: head and neck squamous cell carcinoma, READ: rectal carcinoma, LUSC: lung squamous cell carcinoma, UCEC: uterine corpus endometrial carcinoma, BRCA: breast adenocarcinoma, LUAD: lung adenocarcinoma, OV: ovarian serous carcinoma, KIRC: kidney renal clear cell carcinoma, GBM: glioblastoma multiforme. (**C–F**) Representative immunohistochemical analysis of VHL expression in CRISPR-IUE glioma tumors derived from Daam2-overexpression (**D**) or knockout of Daam2 (**F**) and associated controls. (**G–J**) Representative immunohistochemical analysis of Akt pS473 expression in CRISPR-IUE glioma tumors derived from Daam2-overexpression (**H**) or knockout of Daam2 (**J**) and associated controls. Images are representative of analysis performed on six independent tumors for each experimental condition. (**K**) Immunoblot with antibodies against HIF1α, Glut1, and VEGFA from protein lysates derived from CRISPR-IUE glioma overexpressing Daam2 or control. Scale bars in C and G are 50 nm.

DOI: https://doi.org/10.7554/eLife.31926.010

The following figure supplements are available for figure 4:

**Figure supplement 1.** Daam2 effect on Wnt signaling in glioma and RPPA analysis.

DOI: https://doi.org/10.7554/eLife.31926.011

*Figure 4 continued on next page*

*Figure 4 continued*

**Figure supplement 2.** Correlation between Daam2 and proteins suppressed in glioma.
DOI: https://doi.org/10.7554/eLife.31926.012

**Figure supplement 3.** Manipulation of Daam2 expression influences angiogenesis.
DOI: https://doi.org/10.7554/eLife.31926.013

expression of HIF1α targets Glut1 and VEGFA in the Daam2-GOF tumors, finding that these proteins also demonstrate increased expression (*Figure 4K*). Finally, one consequence of these changes in gene expression is elevated levels of angiogenesis (*Fan et al., 2014*; *Keith and Simon, 2007*; *Semenza, 2004*), which we detected in Daam2-GOF tumors via staining for the endothelial marker, CD31 (*Figure 4—figure supplement 3*). These data, combined with our bioinformatics analysis, indicate that Daam2 suppresses VHL-signaling in malignant glioma.

These observations raise the question of whether the effects of Daam2 on glioma tumorigenesis are mediated through its suppression of VHL expression.

To determine whether an epistatic relationship exists, we overexpressed VHL in the presence of Daam2 overexpression in human GBM cell lines, finding that VHL suppresses the increased rate of cell growth mediated by Daam2-alone (*Figure 5A–C*). Next, we extended these studies to our mouse models of glioma, finding that combined overexpression of VHL with Daam2 similarly suppressed the accelerated rate of tumorigenesis and proliferation mediated by Daam2-alone (*Figure 5D–O*; T-V). Moreover, overexpression of VHL resulted in concordant suppression of pAkt and angiogenesis in these tumors (*Figure 5P–S*; *Figure 4—figure supplement 3*). Put together, these data indicate that Daam2 promotes tumorigenesis by suppressing VHL expression in glioma.

## Daam2 facilitates ubiquitination of VHL protein

Analysis of human cancer data, along with our functional studies in mouse models, strongly suggests that Daam2 expression results in the loss of VHL protein. To determine the mechanism by which Daam2 influences the levels of VHL protein, we next performed a series of immunoprecipitation experiments to evaluate the biochemical relationship between Daam2 and VHL. Using protein lysates extracted from our PB-Ras model of glioma, we found that Daam2 co-immunoprecipitates with VHL (and vice versa), suggesting that these proteins associate (*Figure 6A*).

Given that Daam2 associates with VHL, and expression of VHL protein is negatively correlated with Daam2, we hypothesized that these expression dynamics are the result of Daam2 promoting the degradation of VHL. To test this we co-transfected Daam2 and VHL in 293 cells, and measured the rate of cyclohexamide-mediated degradation, finding that VHL degradation is enhanced in the presence of Daam2 (*Figure 6B,D*). The ubiquitination pathway is a central mechanism that oversees protein degradation, (*Pickart and Eddins, 2004*; *Kerscher et al., 2006*; *Ulrich and Walden, 2010*), therefore, we next examined whether Daam2 facilitates the ubiquitintion of VHL. Using in vitro systems, we found that in the presence of elevated levels of Daam2, the extent of VHL ubiquitination is substantially increased and levels of VHL protein demonstrate a concomitantly decrease (*Figure 6C*). Together, these mechanistic studies reveal that the underlying biochemical relationship between Daam2 and VHL is mediated by ubiquitin-driven protein degradation.

## Discussion

Using factors critical for central nervous system (CNS) development as an entry point to identify new mechanisms that contribute to tumorigenesis, we found that the glial development factor, Daam2 promotes glioma tumorigenesis across human glioma cell lines and multiple mouse models of glioma. Protein screening and bioinformatics studies revealed that Daam2 and VHL expression is inversely correlated across a broad spectrum of cancers, while functional interrogation of this relationship demonstrated that Daam2 promotes tumorigenesis via suppression of VHL expression. Mechanistically, Daam2 associates with VHL and stimulates its ubiquitination and degradation. Together, these studies are the first to define an upstream mechanism regulating VHL protein turnover in cancer and describe the role of Daam2 in tumorigenesis.

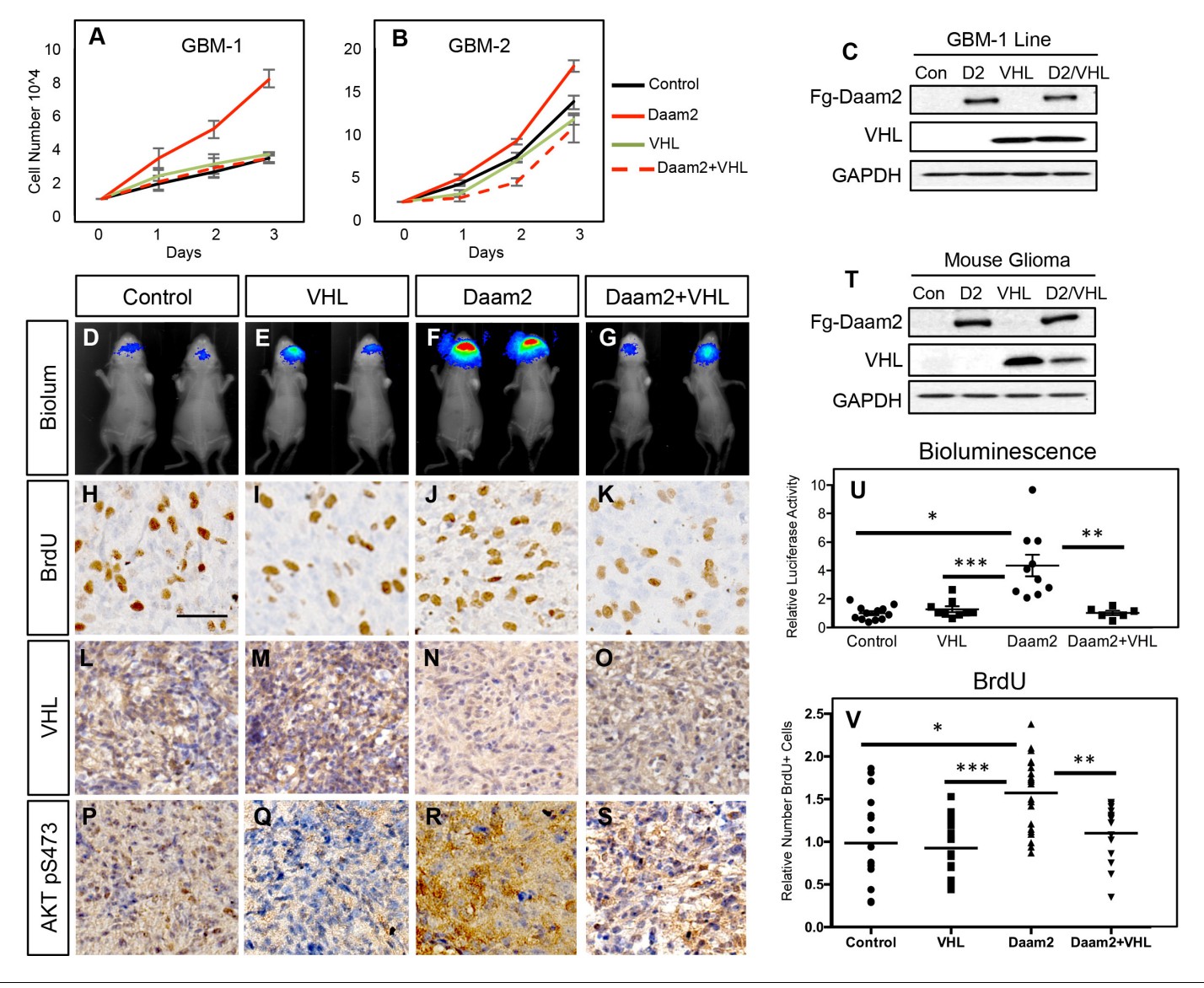

**Figure 5.** Daam2 promotes gliogenesis via suppression of VHL. (**A–B**) Cell proliferation analysis of human GBM1 and GBM2 cell lines infected with lentivirus containing GFP control, Daam2, VHL, or Daam2 +VHL. Associated immunoblots are in C. (**D–G, U**) Representative bioluminescence imaging of mice bearing PB-Ras glioma with Daam2 overexpression, VHL overexpression, Daam2 +VHL overexpression, or control, imaged at 10 days post-natal. Imaging quantification in U is derived from 10 Daam2-overexpression, 9 VHL overexpression, 6 Daam2 +VHL, and 12 control mice; *p=0.0001, **p=0.0046, ***p=0.0016 (**H–K, V**) Immunohistochemistry analysis of BrdU expression from mice bearing PB-Ras glioma from each experimental condition. Relative number of BrdU expressing cells is quantified in V and is derived from six total mice, four slides per mouse, from each experimental condition; *p=0.0006, **p=0.0009, ***p<0.0001. (**L–O**) Representative immunohistochemical analysis of VHL expression in PB-Ras glioma from described experimental conditions and associated controls. (**P–S**) Representative immunohistochemical analysis of Akt pS473 expression in PB-Ras glioma tumors derived from the described experimental conditions and associated controls. (**T**) Representative immunoblot showing overexpression of Daam2 and VHL in the various experimental conditions. Images in L-O are representative of analysis performed on six independent tumors for each experimental condition. Statistics derived by one-way analysis of variance (ANOVA) and followed by Tukey's test for between-group comparisons. Scale bar in F is 50 nm. Error bars in U and V are ± SEM.

DOI: https://doi.org/10.7554/eLife.31926.014

## Daam2 stimulates glioma tumorigenesis

Leveraging existing TCGA expression data across a broad spectrum of malignancies, we found that Daam2 is most highly expressed in glioma and melanoma. These expression characteristics in glioma were confirmed using tissue arrays and functional studies revealed that Daam2 promotes cell

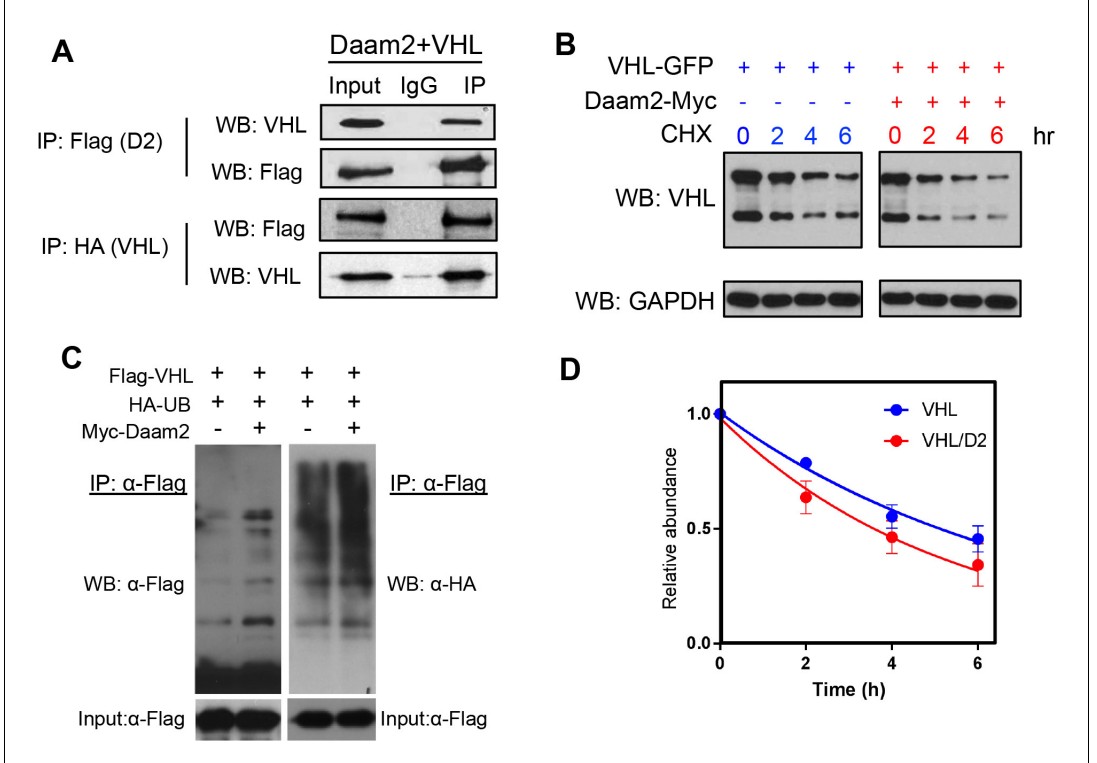

**Figure 6.** Daam2 mediates VHL degradation and ubiquitination. (**A**) Immunoprecipitation of Flag-Daam2, followed by immunoblot with VHL on lysates derived from PB-Ras glioma overexpressing Flag-Daam2. Detection of endogenous VHL in the 'IP' lane, indicates Daam2 associates with VHL in mouse glioma. The reverse IP was also performed using HA-VHL to pulldown Flag-Daam2. (**B,D**) 293 cells were transfected with VHL and Daam2 (or control). 24 hr after transfection, cells were treated with 40 μg/ml cycloheximide (proteasome inhibitor) and incubated for the indicated time before harvest. Western blotting was performed to monitor VHL levels using anti-VHL antibody. Quantification in D shows the effect of Daam2 on VHL stability. Half-life of VHL (5.082 hr) was significantly decreased in the presence of Daam2 (3.679 hr). Error bars represent mean ± SEM (n = 3). Half-life was determined via nonlinear, one-phase exponential decay analysis (half-life parameter, K, is significantly different in two conditions: p=0.0081). Error bars represent mean ± SEM (n = 3). (**C**) Daam2 enhances VHL ubiquitination. 293 T cells were co-transfected with Flag-VHL, HA-UB(ubiquitin), and Myc-Daam2. Cells were treated with MG132 (10 ug/ml) 6 hr before harvest. Whole-cell lysate were immunoprecipitated with anti-Flag M2 beads then analyzed by western blotting with Flag and HA antibodies.

DOI: https://doi.org/10.7554/eLife.31926.015

proliferation and tumorigenesis in human and mouse glioma models. These are the first studies to describe the role of Daam-family proteins in tumorigenesis. Daam1 and Daam2 have highly conserved formin domains yet exhibit non-overlapping expression patterns in the developing CNS (*Habas et al., 2001*; *Kida et al., 2004*; *Lee and Deneen, 2012*; *Nakaya et al., 2004*), suggesting distinct functions during development. Indeed, Daam2 functions in this context via canonical Wnt-signaling, while Daam1 operates via the non-canonical Wnt, planar cell polarity (PCP) pathway (*Habas et al., 2001*; *Lee and Deneen, 2012*; *Li et al., 2011*; *Liu et al., 2008*; *Zhu et al., 2012*). It will be important to discern whether this functional diversity between Daam1 and Daam2 is also applicable to CNS malignancies. Given that the PCP pathway has been implicated in tumor cell invasion and migration (*Anastas and Moon, 2013*; *Paw et al., 2015*; *Weeraratna et al., 2002*), it is possible that Daam-family proteins also contribute to these features of tumorigenesis. Recent studies have implicated Daam1, Daam2 and other formin-family genes in the migration of breast and neuroblastoma cell lines (*Luga et al., 2012*), respectively, suggesting that the Daam-family proteins may also contribute to invasion and metastasis in other malignancies.

Our previous studies have shown that Daam2 potentiates canonical Wnt signaling through the clustering of existing Wnt receptor complexes (*Lee and Deneen, 2012*). While the Wnt pathway has been implicated in several forms of cancer, including medulloblastoma, mutations in key components of the Wnt-pathway and dysregulated Wnt-signaling has not been widely linked to low- or

high-grade glioma (*Bienz and Clevers, 2000*; *de La Coste et al., 1998*; *Klaus and Birchmeier, 2008*; *Lustig and Behrens, 2003*; *Morin et al., 1997*; *Rubinfeld et al., 1997*; *Zurawel et al., 1998*). This, coupled with the fact that Daam2 acts upon existing Wnt receptor complexes, may explain why we did not witness any overt changes in Wnt activity when we manipulated Daam2 expression in our glioma models. Nevertheless, we cannot formally rule out a possible role for Daam2 in canonical or non-canonical Wnt signaling in glioma or in other malignancies driven by Wnt dysregulation.

Finally, our observations that Daam2 expression is increased in a majority of glioma tumors raises the question how it becomes dysregulated. One possibility is that it is actively regulated by transcription factors or microRNAs that are present in glioma. Currently, these mechanisms associated with Daam2 remain undefined and identifying them are key areas of future investigation that will shed new light on how developmental dysregulation influences tumorigenesis. Another possibility to consider is that Daam2 dysregulation is a passive by-product of its expression in a cell lineage that is over-represented in glioma. Daam2 is expressed in both glial lineages (oligodendrocytes and astrocytes), in both precursor and mature stages of development. Both of these lineages are present in one form or another within the bulk tumor, therefore its possible that elevated Daam2 expression is the result of its expression in analogous cell populations that comprise the primary, bulk tumor. Deciphering which scenario (passive or active) is responsible for Daam2 dysregulation in glioma is an important area of future investigation.

## New parallels between development and cancer

Our observation that manipulation of Daam2 expression promoted tumorigenesis, but not canonical Wnt signaling, points to the possibility that it functions through a Wnt-independent mechanism. RPPA and bioinformatics approaches revealed that Daam2 expression is inversely correlated with a cohort of genes (*Figure 4—figure supplement 2*), including VHL, across a broad spectrum of cancers. Moreover, the RPPA data analysis identified a larger cohort of proteins that were downregulated in the presence of Daam2 in our mouse model of glioma (*Figure 4—figure supplement 1*). Thus, future studies may also be geared towards delineating the links between Daam2 and these other downregulated candidates (i.e. Claudin-7 and Syk, etc.) in the context of glioma and glial development. These observations, coupled with our findings that Daam2 associates with- and facilitates- the ubiquitination of VHL, suggest that it may also play a role in the ubiquitination pathway. The nature of this prospective relationship between Daam2 and the existing ubiquitination machinery, and whether these relationships are specific to malignancy or can be extended to development are areas of future investigation and represent potentially new parallels between development and cancer. Indeed dysregulation of ubiquitination is linked to protein aggregation associated with several neurological disorders, functioning through both neuronal and glial populations (*Jansen et al., 2014* and *Hegde and Upadhya, 2007*).

Another feature of Daam2 is that it functions to suppress the differentiation of oligodendrocyte progenitor cells (OPC) during development and after injury (*Lee et al., 2015*). Studies in mouse models suggest that OPCs may serve as a cell of origin for glioma, while NG2-positive OPC populations are endowed with tumor initiating properties (*Ligon et al., 2007*; *Liu et al., 2011*; *Persson et al., 2010*; *Stiles and Rowitch, 2008*; *Sugiarto et al., 2011*; *Yadavilli et al., 2015*). Moreover, OPCs have latent proliferative capacity that is essential for injury responses, suggesting parallels between the processes that drive injury responses and tumorigenesis. Indeed recent studies have linked the VHL-HIF1α signaling axis to OPC development and neonatal hypoxic injury responses (*Yuen et al., 2014*). That Daam2 promotes tumorigenesis via suppression of VHL and also regulates OPC development and injury responses, suggests that it may also utilize these tumorigenic mechanisms in the context of injury. Together these findings add to the emerging evidence that OPC associated signaling networks play critical roles in glioma pathophysiology and further reinforce the parallels between neurological disorders and cancer biology.

## Regulation of VHL in cancer

Amongst the candidates identified in our RPPA screen, we chose to focus our studies on VHL because it's a potent effector of tumorigenesis across many forms of cancer. Our mechanistic studies revealed that Daam2 promotes glioma tumorigenesis via suppression of VHL, a classic tumor suppressor that promotes HIF1α degradation and inhibits Akt activity. Importantly, inactivating

mutations in VHL are predominately found in ccRCC and are very rare in most other forms of cancer, including glioma. These observations raise the critical question of how VHL becomes dysregulated in cancers that do not have these mutations. Our studies define a novel regulatory mechanism that operates upstream of VHL in cancer and provides additional insight into how its expression is extinguished in tumors that do not have inactivating mutations. Moreover, because Daam2-VHL expression demonstrates a robust, inverse correlation across a spectrum of cancers, this is likely to be a generalized mechanism of VHL suppression in cancer. Future areas of investigation include structure/function mapping of the Daam2-VHL domains in the context of ubiquitination, tumorigenesis, and HIF1α signaling.

Interestingly, prior studies in yeast have shown that VHL can be degraded via the Hsp70 and Hsp110 chaperone system (*McClellan et al., 2005*; *Melville et al., 2003*). Given that Hsp's are expressed across a host of cancers (*García-Morales et al., 2007*; *Nylandsted et al., 2002*; *Sauvageot et al., 2009*), it will be important to determine whether Daam2 engages the Hsp chaperone system to facilitate VHL degradation in glioma and other malignancies. Another intriguing possibility is whether VHL can reciprocally degrade Daam2 protein. Given that VHL functions as an E3 ligase and associates with Daam2, this may be another mechanism by which this complex operates. While we did not observe overt decreases in Daam2 protein levels in the presence of VHL (not shown), such a mechanism may be context or cell lineage specific, requiring additional co-factors or environmental conditions (i.e. hypoxia, etc.). Future studies in other cancer models and cell systems will be required to determine if this is the case.

Given its role as the central regulator of HIF1α, surprisingly little is known about the regulatory biology surrounding VHL and its role in glioma formation. Studies in glioma cell lines suggest that ID2 interference with VHL activity can deregulate HIF1α expression and promote tumorigenesis in xenograft models (*Lee et al., 2016*). Interestingly, ID2 has also been linked to the suppression of OPC differentiation via direct regulation of cell cycle kinetics (*Wang et al., 2001*). These observations further reinforce the relationship between OPC development and glioma tumorigenesis and highlight key parallels between ID2 and Daam2 function across these systems, as both genes suppress VHL, inhibit OPC differentiation, and promote glioma tumorigenesis. Because both proteins associated with- and regulate- VHL, albeit via distinct mechanisms, it's possible that these suppressive functions are coordinated and contribute tumorigenesis. From this, a model emerges where ID2 displaces VHL from the ubiquitin complex, which facilitates its association with Daam2 and its subsequent ubiquitination and degradation. It will be important to decipher whether ID2 and Daam2 functions are in fact coordinated, and the extent to which Hsp proteins participate in this mechanism. Finally, understanding whether and how these prospective relationships are also applicable to OPC development and injury responses may also reveal new insights in CNS repair mechanisms.

## Materials and methods

### In situ hybridization on gliomas

To detect the expression of human Daam2, we generated mRNA probe and performed in situ hybridization on human gliomas. Human glioma tissue microarray (US Biomax, GL803b) contains 35 LGG and 40 GBM and five adjacent normal brain tissue, single core per case. To generate human Daam2 probe, pCMV-SPORT6 containing huDaam2 mRNA sequence (BC078153) was purchased from Openbiosystems (EHS1001-9143512). Antisense probe was produced by T7 RNA Polymerase followed by Sal1 enzyme digestion. The sense probe was generated by SP6 RNA Polymerase followed by Not1 enzyme digestion. To apply ISH on human TMA, tissue array was firstly de-paraffinized by Xylene, 100% EtOH, 90% EtOH, 70% EtOH, 50% EtOH, two times of 5 min for each step. Then deproteinization was conducted by proteinase K (10 ug/ml) for 5 min, 4% PFA for 15 min, 0.25% acetic anhydride in 0.1M triethanolamine for 5 min, washing with PBS in between for 5 min. Hybridization of the RNA probe at 65°C for overnight after pre-hybridization at 65°C for 1 hr. Chromogenic detection was finished the following day. Three times 2xSCC washing at 65°d for 15 min each first, then rinsing with PBT, 1 hr blocking (20% lamb serum in PBT), 2 hr antibody incubation (DIG-AP, 1:2000), washing with PBT, and 10 min AP blocking. The slide was developing in NBT/BCIP in the dark for overnight. The reaction was stopped by AP buffer once the signal was developed.

The intensity grades (0, 1, 2, 3) is scored by the pathologist, Dr. Carrie Mohila. To detect the expression of Daam2 in mouse gliomas, we used mRNA probes (*Lee et al., 2015*)

## Mice

To generate mouse gliomas, we used pregnant wild type and Daam2 knockout mice for IUE surgeries. The pregnant wild-type mice were purchased from The Center for Comparative Medicine (CCM) at Baylor College of Medicine (BCM). The Daam2 knockout mice were generated previous studies (*Lee et al., 2015*). We also purchased SCID (Taconic) mice for human glioma transplantation assay. All procedures were approved by the Institutional Animal Care and Use Committee at Baylor College of Medicine and conform to the US Public Health Service Policy on Human Care and Use of Laboratory Animals.

## Constructs, shRNAi lentivirus, cloning

### Lentivirus constructs

For in vitro functional assays in human GBM cell lines, ectopic Flag-tagged Daam2 was cloned into FUIGW plasmid, HA-tagged VHL was cloned into pHage-puro Gateway vector. Human Daam2 shRNAi were purchased from TRC Lentiviral Library, Dharmacon. Mature Antisense- ATGC TCCGGAGGAAATTTAGC was used as the control. The combination of mature Antisense- A TTTAGCCGGTTTATTGCCCG and mature Antisense- ATCAAACGCGTACAGACTCTG was used as the Daam2-shRNAi.

### PiggyBac constructs

For PB-Ras glioma model, Flag-tagged Daam2, HA-tagged VHL, and Luciferase were cloned into the pCAG vector. pbCAG-GFP-2a-HRas, pbCAG-GFP were a gift from Dr. LoTurco. For CRISPR glioma model, oligo primers were used to clone sgRNA targeting *Nf1, Trp53, and Pten* into the pX330-Cas9 vector. p53: 5'-CACCGCCTCGAGCTCCCTCTGAGCC-3', PTEN: 5'- CACCGAGATCGTTAG-CAGAAACAAA-3', NF1: 5'-CACCGCAGATGAGCCGCCACATCGA-3'

### Expression constructs

Flag-tagged Daam2, Myc-tagged Daam2, HA-tagged VHL, Flag-tagged VHL, GFP-tagged VHL were cloned into pcDNA, pCS2, pcDNA, p3xFLAG-CMV-9, pmaxFPTM-Green-N, respectively. HA-Ub was gift from Dr. Yoo.

### Lentivirus generation

Lentivirus production was carried out by transfecting FUIGW backbone with Gagpol and VSVG plasmids into 293 T cells in a 15 cm dish using TransIT-293 reagent (Mirus2705. pHage and shRNAi plasmids are transfected with pMD2.G and dr8.2 plasmids. After transfection, lentivirus supernatant was collected twice over 48 hr and ultra-centrifuging at 25,000 rpm for 2 hr. Purified and concentrated virus was diluted with DMEM media.

## Cell growth curve and agar assays

### Growth curve

For in vitro human GBM cell line studies, adult GBM-1 and GBM-2 cell lines were kindly provided from Dr. Nabil Ahmed (*Hegde et al., 2013*) (*Ahmed et al., 2010*). These cell lines were generated from freshly resected human GBM samples as described in the (*Hegde et al., 2013*) and *Ahmed et al., 2010*) and whole transcriptome sequencing confirmed that they are of human origin and contain gene expression patterns consiste with GBM. In addition, these cell lines tested negative for mycoplasma. To manipulate the expression of Daam2 in GBM cell lines, we used the lentivirus as described above. To virally infect human primary GBM cells, $1.5 \times 10^4$ cells were plated in 6-well plates with DMEM +10% FBS. Cells were infected with either Daam2 or Daam2 shRNAi virus for 14 hr to make the stable cell line for GOF and LOF study, respectively. Stable Daam2 over-expression (GFP$^+$) cell lines were sorted via FACS. Stable Daam2 shRNAi cell lines were selected by puromycin. To test the relationship between Daam2 and VHL, Daam2 over-expression cell line was infected with VHL virus. To assess rates of cell growth, $2 \times 10^4$ cells were plated in 12-well plates and cells were

counted over the course of three days. At each time point, cells were detached by Trypsin and calculated using TC20 Automated Cell Counter (Bio-Rad).

## Agar assay
For the agar assay, $2.5 \times 10^4$ cells (cell agar layer) was mixed with 5 ml Iscove's 1.4% Nobel Agar + 40% FBS, covered by top and bottom coating agar layers (2 ml Iscove's 1.4% Nobel Agar + 20% FBS) in a 6 mm plate. We tightly monitored the colony formation for several weeks.

## qRT-PCR
$5 \times 10^6$ cells were cultured in 6-well plates. RNA was extracted using RNeasy Plus Mini Kit (Qiagen, #74134) according to the protocol. 1000 ng RNA was used for reverse transcription into cDNA with a total volume of 20 ul. cDNA was diluted into 100 ul with Nuclease-free water. 5.75 ul cDNA was mixed with SYBR reagent and primers for running qRT-PCR program. The following primers were used: Daam2 Left 5'-caaagcccaaagtggaagc; Daam2 Right 5'-catctgtctaagacgcttgctg.

## In utero electroporation (IUE)
To generate mouse gliomas, we performed in utero electroporation (IUE). Briefly, the uterine horns were exposed, and DNA combination was injected into the embryonic lateral ventricles along with Fast Green dye as the indicator. Then electroporation was accomplished by BTW tweezertrodes connected with the pulse generator (BTX 8300) in the setting of 33V, 55 ms per pulse for six times, 100 ms intervals. In CRISPR-IUE model, the DNA combination is composed of the 'helper-plasmid' pGLAST-PBase (2.0 ug/ul) and all the other DNA (1.0 ug/ul), including pbCAG-GFP, pbCAG-Luciferase, crNF1, crPTEN, crp53 (*Chen and LoTurco, 2012*; *John Lin et al., 2017*). In the HRas-IUE model, the DNA combination is composed of pGLAST-PBase (2.0 ug/ul) and others (1.0 ug/ul), including pbCAG-GFP2aHRas and pbCAG-Luciferase. For the IUE/Daam2/VHL overexpression study, pbCAG-FlagDaam2 (1.0 ug/ul), pbCAG-HA-VHL (1.0 ug/ul) were co-injected with the CRISPR-IUE or the Ras-IUE master mix described above.

## Glioma xenograft assays
Six-week-old SCID male mice (Taconic) were used for human GBM cell line transplantation. $4 \times 10^4$ luciferase-infected primary GBM cells were injected into each mouse brain, at the location of the 1 mm front, 2 mm right, 3 mm deep from the Bregma. Animals were euthanized and perfused at six weeks after transplantation of GBM cells. Brains were fixed in 4% paraformaldehyde and 70% EtOH overnight, respectively. After fixation, brains were embedded in paraffin, sectioned and subjected to molecular and pathological analysis via immunostaining or hematoxylin and eosin staining.

## Bioluminescent imaging and analysis
To measure the tumor growth after manipulation of Daam2, mice are subjected to bioluminescent imaging before harvesting. Mice were monitored once a week. D-Luciferin (Perkin Elmer, #122799) was diluted to 15 mg/ml with PBS and injected into each mouse at a dose of 10 ul/g body weight. After 10 min, mice were placed inside a Bruker FX Pro Imager for 2 min Bioluminescence Imaging and 10 s X-ray Imaging. X-ray image was transparently overlapped with the bioluminescence image. A free circle surrounded the region of interest (ROI) was selected for the quantification of the luciferase intensity. Relative luciferase intensity was measured as the intensity of control group was normalized as 1; experiment group = actual value of experiment group/the actual value of control group.

## BrdU assay
To evaluate tumor proliferation, mice were subjected intraperitoneal injection with BrdU (200 µg/g body weight) 4 hr before harvesting. Brains were perfused and fixed overnight in 4% paraformaldehyde and 70% EtOH overnight for paraffin embedding. For the BrdU immunohistochemistry, additional 2N HCl treatment was performed.

## Tumor dissection and dissociation
Tumors derived from either IUE or transplantation were dissected from whole mouse brains. Tumor regions were visualized as GFP positive tissue under a fluorescence dissection microscope. For

paraffin sections, the brain was fixed in 4% Formalin then 70% Ethanol overnight. For western blot and RPPA, GFP-positive tissues were dissociated from the whole brain then homogenized with lysis buffer.

## Hematoxylin and eosin stain (H&E), Immunohistochemistry (IHC), and Antibodies

For H&E staining, slides were firstly deparaffinized by sequentially three times of 3 min Xylene, 100% EtOH, 95% EtOH, 80% EtOH, 70% EtOH. Hematoxylin was stained for 5 min, then rinsing with tap water till no more color changing. 2–3 dips in the acidic solution (1% HCl in 70% EtOH) reduced the background. After rinsing with tap water, Eosin was staining for 1 min, following by tap water rinsing. Finally, dehydration was accomplished by three times of 5 min 95% EtOH, 100% EtOH, and Xylene.

For IHC, deparaffinization process is the same as H&E staining. Antigen retrieval was performed by 10 min microwaving using Na-Citrate pH6.0. Endogenous peroxidases were blocked using 3% $H_2O_2$. After 1 hr serum blocking, slides were incubated with antibody overnight in cold room. The other day, slides were rinsed off in PBS and incubated with secondary antibody for 1 hr. Then DAB and hematoxylin were applied for the color matrix and counterstain. Finally, dehydration was the same as the H&E process. CD31 staining was quantified using approaches described in *Tian et al. (2017)*.

The following antibodies were used for IHC and western blot: BrdU (anti-rat, Abcam, ab6326, 1:200), VHL (anti-rabbit, Santa Cruz, sc5575, 1:20), VHL (anti-mouse, #556347, 1:500), CD31 (anti-rabbit, Abcam, ab28364, 1:100), pAKT (Ser473) (anti-rabbit, Cell Signaling, #9271, 1:1000), HIF1a (anti-mouse, Novus, NB100-105, 1:500), Flag (anti-mouse, Sigma, F1804, 1:1000), HA (anti-rat, Roche, clone 3F10, 1:500), GAPDH (anti-mouse, Millipore, AB2302, 1:1000), and LacZ (anti-rabbit, Cappel, 55976, 1:1000).

## Reverse phase protein array (RPPA)

GFP$^+$ tumor tissues were dissociated under a fluorescence dissection microscope and sorted with FACS machine. Tumor cells were homogenized with ice-cold lysis buffer. Supernatants (tumor lysates) were transferred after centrifuging at 4°C, 14,000 rpm for 10 min. The lysates were mixed with 4X SDS sample buffer. Samples were boiled for 5 min at a final protein concentration of 1–1.5 ug/ul and a total volume of 40 ul. The lysis buffer was composed of 1% Triton X-100, 50 mM HEPES, pH 7.4, 150 mM NaCl, 1.5 mM MgCl2, 1 mM EGTA, 100 mM NaF, 10 mM Na pyrophosphate, 1 mM Na3VO4, 10% glycerol, and contained freshly added protease and phosphatase inhibitors from Roche (# 05056489001 and 04906837001, respectively). Samples were probed with 287 antibodies. The RPPA was performed and analyzed by Functional Proteomics RPPA Core Facility at MD Anderson Cancer Center.

## Immunoprecipitation and in vitro degradation assay

### Immunoprecipitation

To test the biochemical association between Daam2 and VHL, we used 293 T cells for co-immunoprecipitation assay. $2 \times 10^5$ cells were plated into the six well plate one day before transfection. 500 ng Flag-tagged Daam2 and HA-tagged VHL was transfected by iMFectin DNA Transfection Reagent (GenDEPOT) according to the protocol. Cells were harvested by cold PBS and spin down by centrifuging at 3000 rpm, 5 min. The cell pellet then was lysis by 1 ml lysate buffer (20 mM HEPES pH7.9, 100 mM NaCl, 0.5% TritonX100, 1 mM EDTA, 5% Glycerol) with protease inhibitor (Roche). To fully lysis the cells, cells were subjected 10 s sonication and three times of 10 s vortex. Then cell lysate (supernatant) was collected after 15 min, 13,000 rpm centrifuge. 30 ul samples were collected as Input; the other cell lysate was subjected for overnight IP by adding 12 ul M2 FLAG agarose (Sigma). IP samples were washed with three times cell lysate buffer followed by 13,000 rpm 5 min to spin down and separate the agarose beads. 30 ul 2xSDS loading buffer was added to individual samples, 10 ul of samples were loaded for Western blot. Flag (anti-mouse, Sigma, F1804, 1:1000), VHL (anti-mouse, #556347, 1:500), HA (anti-rat, Roche, clone 3F10, 1:500), GAPDH (anti-mouse, Millipore, AB2302, 1:1000)

For the in vitro ubiquitin assay, 100 ng Myc-tagged Daam2, 500 ng Flag-tagged VHL, and 600 ng HA-tagged Ub were transfected into six-well plate. 24 hr post- transfection, cells were treated with 40 ug/ml MG132 (proteasome inhibitor) for 6 hr. Lysate buffer for Ubiquitin contained additional de-Ubiquitin reagent (50 ug/ml N-Ethylmaleimide, 75 ug/ml 1,10-Phenanthroline).

### In vitro degradation assay

To test VHL degradation, $1 \times 10^5$ 293 T cells were plated into the 12 well plate one day before transfection. 100 ng Myc-tagged Daam2 and GFP-tagged VHL was transfected by iMFectin DNA Transfection Reagent (GenDEPOT) according to the protocol. 30 hr post-transfection, cells were treated with 50 ug/ml CHX for 0, 2, 4, 6 hr respectively. Lysis and Western blot process is the same as previously described. To quantify the protein abundance, the western blot band intensity in 0 hr sample of both VHL -/+D2 groups are set as '1', then the intensity of 2, 4, 6 hr degradation western blot band is normalized and plotted using Nonlinear Regression Curve by GraphPad.

### TCGA database analysis and comparative bioinformatics

RNA expression data underlying the results presented in *Figure 1* were generated by TCGA Research Network (http://cancergenome.nih.gov/). All data used in this study were publicly available, e.g. from The Broad Institute's Firehose pipeline (http://gdac.broadinstitute.org/), doi:10.7908/C11G0KM9. From TCGA, we collected molecular data on 10224 tumors of various histological subtypes (ACC project, n = 79; BLCA, n = 408; BRCA, n = 1095; CESC, n = 304; CHOL, n = 36; COAD/READ, n = 623; DLBC, n = 48; GBM, n = 161; HNSC, n = 520; KICH, n = 66; KIRC, n = 533; KIRP, n = 290; LAML, n = 173; LGG, n = 516; LIHC, n = 371; LUAD, n = 515; LUSC, n = 501; MESO, n = 87; OV, n = 262; PAAD, n = 178; PCPG, n = 179; PRAD, n = 497; SARC, n = 259; SKCM, n = 469; TGCT, n = 150; THCA, n = 503; THYM, n = 120; UCEC, n = 545; UCS, n = 57; UVM, n = 80) from TCGA, for which RNA-seq data (v2 platform alignment) were available. Correlation between VHL and AKT pS473 protein on TCGA samples, were obtained from The Cancer Proteome Atlas, or TCPA, using level four data from the portal (*Akbani et al., 2015*; *Li et al., 2013*).

### Statistical analysis

One-way ANOVA was used to analyze bioluminescent intensity and BrdU-positive cell counts to determine the differences between Ctrl, D2 and D2/VHL groups, followed by Tukey's test to compare between individual groups, which is demarcated by an asterisk in the graphs. Independent *t*-test was used to analyze the differences in bioluminescent intensity and BrdU-positive cell counts between Ctrl vs. D2, Het vs. KO, Scrambled-shRNAi vs. D2-shRNAi.

## Acknowledgements

This work was supported by grants from the Sontag Foundation (BD), Cancer Prevention Research Institute of Texas (RP150334 and RP160192 to BD and CC), and the National Institutes of Health (NS071153 and NS089366 to BD). National Multiple Sclerosis Scoiety (RG 1508-08406 to HKL). We acknowledge the assistance of the Baylor College of Medicine Mouse Phenotyping Core, the Lester and Sue Smith Breast Center's Pathology Core, and Functional Proteomics RPPA Core Facility at MD Anderson Cancer Center, this facility is funded by NCI # CA16672. This project was also supported in part by the IDDRC grant number 1U54 HD083092 from the Eunice Kennedy Shriver National Institute of Child Health and Human Development.

## Additional information

### Funding

| Funder | Grant reference number | Author |
|---|---|---|
| National Multiple Sclerosis Society | RG 1508-08406 | Hyun Kyoung Lee |
| Cancer Prevention and Research Institute of Texas | RP16019 | Benjamin Deneen |

| National Institutes of Health | NS071153 | Benjamin Deneen |
|---|---|---|
| Cancer Prevention and Research Institute of Texas | RP150334 | Benjamin Deneen |
| National Institutes of Health | NS089366 | Benjamin Deneen |

The funders had no role in study design, data collection and interpretation, or the decision to submit the work for publication.

## Author contributions

Wenyi Zhu, Conceptualization, Data curation, Formal analysis, Supervision, Validation, Investigation, Methodology, Writing—original draft, Writing—review and editing, Data analysis; Saritha Krishna, Formal analysis, Investigation, Methodology, Project administration; Cristina Garcia, Bartley D Mitchell, Investigation, Methodology; Chia-Ching John Lin, Formal analysis, Investigation, Methodology; Kenneth L Scott, Formal analysis, Investigation, Key reagents; Carrie A Mohila, Investigation, Methodology, Data analysis; Chad J Creighton, Data curation, Formal analysis, Investigation, Data analysis; Seung-Hee Yoo, Investigation, Methodology, Key reagents; Hyun Kyoung Lee, Conceptualization, Resources, Data curation, Formal analysis, Supervision, Funding acquisition, Validation, Investigation, Methodology, Writing—original draft, Writing—review and editing, Data analysis; Benjamin Deneen, Conceptualization, Funding acquisition, Validation, Investigation, Visualization, Methodology, Writing—original draft, Project administration, Writing—review and editing, Data analysis

## Author ORCIDs

Benjamin Deneen (iD) http://orcid.org/0000-0002-6335-1081

## Ethics

Animal experimentation: This study was performed in strict accordance with the recommendations in the Guide for the Care and Use of Laboratory Animals of the National Institutes of Health. All of the animals were handled according to approved institutional animal care and use committee (IACUC) protocol AN-5162 and AN-6100 at the Baylor College of Medicine. All surgery was performed under isoflurane anesthesia, and every effort was made to minimize suffering.

## Decision letter and Author response

Decision letter https://doi.org/10.7554/eLife.31926.018
Author response https://doi.org/10.7554/eLife.31926.0219

# Additional files

## Supplementary files

• Supplementary file 1. RPPA data analysis.
DOI: https://doi.org/10.7554/eLife.31926.016

• Transparent reporting form
DOI: https://doi.org/10.7554/eLife.31926.017

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
