## [Decision Letter]

Thank you for choosing to send your work, "Daam2 Driven Degradation of VHL Promotes Gliomagenesis", for consideration at *eLife*. Your initial submission has been assessed by a Senior Editor and three reviewers. Although the work is of interest, we regret to inform you that the findings at this stage are too preliminary for further consideration at *eLife*.

Specifically, the reviewers felt that the manuscript was interesting but also raised many concerns that would require substantial revisions. These include better demonstration of Daam2 over-expression, confirmation of Daam2 loss of function, better descriptions of the glioblastoma lines, etc. I refer you to the reviews below for details.

Reviewer #1:

The report by Deneen and colleagues reports that coordinate regulation of Daam2 and VHL is observed in many cancers, and that Daam2 expression functions to increase tumorigenicity through suppression of VHL levels (ubiquitin-mediated degradation). The experiments are well executed and convincing.

How quantitative is the DAAM2 expression described in Figure 1? It would be preferable to look at protein expression in tissue microarrays or perform real time quantitative RT-PCR for these analyses. In addition, it would be useful to compare normal tissue from the exact same location as the tumor whenever possible. This could be performed using mouse gliomas, and should be included to strengthen the importance to gliomas. For example, in the Barres atlas (http://web.stanford.edu/group/barres_lab/brain_rnaseq.html), it appears as though Daam2 is robustly expressed in normal astrocytes.

What the developmental expression pattern of Daam2 in the mouse or human? Is it inversely correlated with VHL expression?

Does Daam2 overexpression increase proliferation or decrease apoptosis?

In the analysis shown in Figure 4—figure supplement 2, VHL was not the top candidate. How did the authors exclude the other compelling candidates, and why did they focus only on VHL?

In the TCGA datasets, is there a strong correlation between DAAM2 expression and HIF-1a expression?

One of the key points of the report is that DAAM2 requires VHL. While they demonstrate sufficiency, the necessity portion of this dependency is not as clearly revealed. What is the effect of VHL expression alone in Figure 5? One elegant way to demonstrate the dependency is to show that mutations that impair VHL-DAAM2 binding reverse the ability of DAAM2 to accelerate glioma growth in vitro and in vivo.

Previous studies from the Deneen laboratory suggest that DAAM2 regulates Wnt signaling in oligodendrocytes. Was this pathway excluded in the context of tumorigenesis?

Reviewer #2:

In the current manuscript, entitled "Daam2 Driven Degradation of VHL Promotes Gliomagenesis," Zhu and colleagues argue that Daam2 drives glioblastoma development in human and mouse models through a novel regulation of the VHL tumor suppressor. Daam2 overexpression in cells and mice associates with increased cell and tumor growth, while Daam2 depletion in glioma models significantly impairs tumor growth. While Daam2 has been reported to regulate the Wnt pathway by these investigators, they found a lack of association of Daam2 and Wnt here. Thus omic and protein array analyses led them to discover an inverse correlation of Daam2 expression and VHL. Accordingly, Daam2 expression induces HIF1 stabilization and Akt phosphorylation, two mechanistic consequences of VHL loss. The investigators go on to show that Daam2 plays a direct role in the regulation of VHL through a protein protein interaction and a consequent ubiquitylation of VHL. Together the studies identify a new and exciting role for Daam2 in glioma, and a new mechanism of VHL downregulation in cancer, which would have broad implications. The manuscript could be improved by a number of edits.

1) This is not the first study to show an upstream mechanism of VHL regulation in cancer. The authors correctly point out in their Discussion that ID2 has been shown to regulate VHL, there is also a miRNA story in the literature (Li, et al. 2017). Thus while these other observations do not take away from the story here, this point is overstated several times in the text and should be corrected. Likewise, a comment in the Discussion implies that absent of other alterations, it is unclear how VHL function is inactivated in cancer, and the simple answer is tumor hypoxia.

2) Figure 1; high resolution pictures of Daam2 staining that are in the supplemental data should be included in the figure.

3) Better descriptions of the two GBM cell lines would be useful. Do these contain similar driver mutations as the mouse models? Also, quantification of the colony assays would be helpful, rather than simply a representative figure. Do the control cells form colonies at all? Given that they form tumors, one would expect so.

4) Quantification of IHC for example for CD31 would be more convincing than a single picture.

5) HIF elevation in Figure 4 begs the question as to whether HIF target genes are also elevated in cells with altered Daam2.

6) Can the authors speculate as to how Daam2 is functioning to facilitate VHL ubiquitylation? Is it an E3 ligase? Is there a domain structure of Daam2 that would support a hypothesis?

Reviewer #3:

The Von Hippel-lindau protein is a tumor suppressor that, through conventional loss-of-function (i.e. germline deletion of one copy followed by somatic l.o.h.) gives rise to cancers of liver, kidney and vagina. In this manuscript, Zhu et al. make the provocative claim that Von Hippel-lindau (VHL) protein may be attenuated in a non-genetic way and thus contribute to a broader array of cancers than hitherto believed including malignant gliomas.

The point of departure for their work is an in silico survey showing an inverse correlation between expression of the glial developmental factor Daam2 and VHL in a wide variety of cancers including glioma. The authors go on to a set of functional studies with human glioma cell lines and mouse glioma models to show that Daam2 promotes glioma growth by suppressing VHL expression. Mechanistically, the claim is that Daam2 associates with VHL and thereby facilitates its ubiquitination and degradation.

A broad body of labor-intensive and technically challenging work is summarized here. The manuscript is well written. The conclusions would, in principle, be of interest to a broad readership in neuro-oncology and in other fields such as tumor metabolism and molecular therapeutics. Unfortunately, some of the data in this manuscript are, at present, inadequate to support the central claims. My concerns are as follows:

1) Figure 2. The claim of this figure is that Daam2 overexpression accelerates the growth of human glioma cell lines. However, the authors do not actually show that Daam2 is overexpressed. They should provide a western blot towards this end. It is actually a bit curious that infection with lentivirus Daam2 has such a pronounced effect since the authors show us (in Figure 1) that gliomas already have high Daam2.

2) Figure 3. An important claim of this figure is that Daam2 loss-of-function impairs growth of human glioma cells. The loss-of-function was achieved via shRNAi-mediated knockdown but there are two problems with the data. First, there is no western blot to show that Daam2 actually was knocked down. Second, there are growing concerns about off-target effects of shRNAi. The authors need a "rescue" experiment with Daam2 construct that is immune to their shRNAi to rule out off-target toxicity.

3) Figure 4—figure supplement 3. The central claim of this figure is that angiogenesis, as marked by CD31, is altered in response to Daam2 levels. The immunostain images shown are not convincing. The authors need to quantify these data.

4) Figure 5. The central claim of these two figures is that ectopic VHL can overcome the accelerated growth induced by overexpression of Daam2 alone. As per comment #1 above, we need western blots – in this case for both VHL and Daam2. It would be interesting to see if the relationship between Daam2 and VHL is symmetric. If high Daam2 suppresses VHL, does high VHL suppress Daam2?

5) Figure 6. The central claim of this figure is that Daam2 forms a complex with VHL thereby promoting ubiquitination and degradation of VHL. These are very important mechanistic claims. I have some doubts about these data. These are:

a) The antibody pulldown experiment needs to be shored up to rule out an adventitious interaction between Daam2 and VHL. Can the pulldown be reversed? Will IP VHL bring down Daam2? Is there any independent evidence of a VHL:Daam2 interaction, e.g. yeast two hybrid trap or proximity ligation assay?

b) Figure 6. The central claim here is that VHL degradation is "significantly enhanced" in the presence of Daam2. I disagree with the descriptor of "significant". The offset in VHL T_1/2_ (5.08hr vs 3.68hr) might be "statistically significant" as the authors state but the overall difference is nuanced and does not seem adequate to account for the striking differential in VHL immunostain as shown in Figure 4. Author's comments?

Overshadowing all of these concerns with the data is a more fundamental question about the role of Daam2 as a non-genetic driver of gliomagenesis. What is the mechanism of leads to high expression of Daam2 in gliomas (Figure 1). Is Daam2 expressed at hyperphysiologic levels in these tumors or, alternatively, do some gliomas simply arise from Daam2-positive cells? If the latter is true than Daam2 is only permissive for gliomagenesis (along with many other things). If the former is true, then we need to know the mechanism.

---

## [Author Response]

Reviewer #1:[…] How quantitative is the DAAM2 expression described in Figure 1? It would be preferable to look at protein expression in tissue microarrays or perform real time quantitative RT-PCR for these analyses. In addition, it would be useful to compare normal tissue from the exact same location as the tumor whenever possible. This could be performed using mouse gliomas, and should be included to strengthen the importance to gliomas. For example, in the Barres atlas (http://web.stanford.edu/group/barres_lab/brain_rnaseq.html), it appears as though Daam2 is robustly expressed in normal astrocytes.

We totally agree with the reviewer’s suggestion. Unfortunately, the Daam2 antibodies at our disposal do not adequately stain human tissues, so we cannot perform the suggested immunostaining experiments. Because of this, we turned to qRT-PCR to assess expression of Daam2 in freshly resected human malignant glioma and adjacent white matter samples. RNA was isolated from the primary tumor and the associated, non-tumor white matter and we performed qRT-PCR, finding that Daam2 expression in glioma was elevated in three of the five samples we queried. Please see Figure 1—figure supplement 1. That Daam2 exhibits heterogeneous expression across 5 glioma samples is consistent with the heterogeneous expression we found in the tissue microarrays and the TCGA analysis (TCGA – see Figure 1; Tissue Array now in Figure 1—figure supplement 1). We noted these parallels in the last paragraph of the subsection “Daam2 is expressed in human and mouse glioma”.

In addition, we assessed Daam2 expression in our mouse model of glioma, comparing tumor and non-tumor expression, finding that these models exhibit elevated levels of Daam2 expression in glioma compared to cortex. Please see Figure 2—figure supplement 1.

What the developmental expression pattern of Daam2 in the mouse or human? Is it inversely correlated with VHL expression?

These are critical questions, that are directly relevant to our studies and we appreciate the reviewer’s consideration of these experiments. Developmental expression of Daam2 in the mouse spinal cord can be found in Lee, et al. Neuron 2015. We also assessed expression in the post-natal and adult cortex, finding that Daam2 is also expressed in GFAP expressing astrocyte populations in both the developing and adult cortex. Please see Figure 2—figure supplement 1.

The issue of Daam2/VHL expression correlation in the normal brain is also a very important issue brought to light by the reviewer. We have stained adult cortex from *Daam2^LacZ/+^* and *Daam2^LacZ/LacZ^* mice with β-gal and VHL antibodies, finding that VHL expression is significantly increased in *Daam2^LacZ/LacZ^* cells in the adult cortex. Please see Figure 4—figure supplement 2 and subsection “Daam2 promotes tumorigenesis through VHL”, first paragraph.

Please note that in all of these experiments, we used a LacZ-knockin allele because we do not have any Daam2-antibodies that adequately stain brain or spinal cord tissue.

Does Daam2 overexpression increase proliferation or decrease apoptosis?

Indeed, increases in tumorigenesis can be engendered by enhanced proliferation, a lack of cell death, or a combination of both. It’s critical to characterize these features of our tumors. To measure proliferation we injected the tumor bearing mice with BrdU prior to harvesting; the extent of BrdU labeling was quantified via immunohistochemistry. These data were included in the original submission and can be found in Figure 2.

To assess changes in cell death, we stained these same tumors for Caspase3 and did not detect any changes in Caspase3 expression. Please see Author response image 1.

**Author response image 1. respfig1:** Caspase3 staining of CRISPR-IUE tumors overexpressing Daam2.

In the analysis shown in Figure 4—figure supplement 2, VHL was not the top candidate. How did the authors exclude the other compelling candidates, and why did they focus only on VHL?

We appreciate the reviewers comment, as explaining the logic underlying the selection of VHL is critical for a comprehensive understanding of our manuscript. As the reviewer can probably appreciate, selection criteria can be subjective and indeed, that is part of our explanation. VHL was selected, in part, because its role in glioma biology was not well understood and linking it to Daam2 might provide additional and novel insight into its role in this context. While VHL was not the top candidate, it did have the third highest correlation score and a very significant p-value. Moreover, amongst the top candidates, VHL is the most potent effector of tumorigenesis and linking Daam2 with a central regulator of tumor growth made the most sense to us. With that said, future studies will be geared towards delineating the links between the other candidates (Claudin-7 and Syk, etc.) and Daam2 in the context of glioma and glial development. We have amended the Discussion to include these points. Please see subsection “New Parallels Between Development and Cancer”, first paragraph and subsection “Regulation of VHL in cancer”, first paragraph.

Another component of our selection criteria is a previous publication showing that VHL promotes oligodendrocyte differentiation (Yuen, et al. Cell 2014). This role for VHL is in contrast to our previous findings that Daam2 suppresses oligodendrocyte development (Lee, et al. Neuron 2015). Based on these opponent roles for VHL and Daam2 in glial development, and that their expression is inversely correlated in glioma, we speculated that they may have an antagonistic relationship that influences glioma tumorigenesis.

In the TCGA datasets, is there a strong correlation between DAAM2 expression and HIF-1a expression?

We appreciate this question, as it’s very important to understand the correlations between Daam2 and the downstream effectors of VHL. This correlation analysis was included in the original submission and can be found in Figure 4. In addition to a strong correlation with HIF1a in the human TCGA datasets, we also found that two key HIF1a targets (VEGFA and Glut1) are upregulated in tumors that overexpress Daam2. Please see Figure 4. Together, these data strongly support the notion that Daam2 overexpression results in dysregulated VHL-HIF1a signaling.

One of the key points of the report is that DAAM2 requires VHL. While they demonstrate sufficiency, the necessity portion of this dependency is not as clearly revealed. What is the effect of VHL expression alone in Figure 5? One elegant way to demonstrate the dependency is to show that mutations that impair VHL-DAAM2 binding reverse the ability of DAAM2 to accelerate glioma growth in vitro and in vivo.

We thank the reviewer for pointing this out, as the effects of VHL manipulation alone are also an important part of this experiment. In Figure 5, we have included VHL overexpression controls in our human glioma cell line and mouse PB-Ras glioma model experiments. As shown in Figure 5, overexpression of VHL alone did not affect growth of the human glioma cell lines or tumorigenesis in our mouse models.

In an effort to test the sufficiency of the Daam2/VHL relationship, we sought to perform a double LOF experiment by deleting VHL via CRISPR in our mouse glioma model generated in Daam2-KO mice. Towards this we first tested the effects of VHL deletion on tumorigenesis in our CRISPR-IUE model, finding that deletion of VHL resulted in a dramatic acceleration of tumorigenesis, please see Author response image 2. Average survival for mice bearing CRISPR-IUE tumors is ~90 days, mice that have been electroporated with guide RNAs that target VHL (plus the NF1/p53/PTEN combination) survive 14-21 days. Moreover, glioma growth is typically detected by bioluminescence at P40-P45, while tumors that lack VHL are detectable as early as P8. Given that deletion of VHL has such a profound effect of tumorigenesis in our mouse models, we are not able to use these models to examine the sufficiency of the Daam2/VHL relationship.

**Author response image 2. respfig2:** CRISPR-IUE tumor model, combined with CRISPR deletion of VHL, generated in a Daam2+/- or Daam2-/- background. Deletion of VHL accelerates glioma formation in this model. (**A**) Kaplan-Meyer survival curve. (C-D) Bioluminescence imaging of tumors at P8.

Given the growth dynamics of tumors that lack VHL, the reviewer’s suggestion to identify mutants that impair Daam2/VHL association is an excellent alternative way to investigate sufficiency. To address this we performed a series of co-IP experiments, testing the association between a series of Daam2 mutants and VHL. As shown in Author response image 3, all the Daam2 truncation mutants at our disposal (see Lee, et al.Developmental Cell 2012) are able to associate with VHL. Because, at this time, we are unable to identify a Daam2-mutant that does not interact with VHL, we are unable to further explore this potentially important mechanism. Based on these results, these experiments appear to be beyond the scope of this study, as they will require additional deletion mapping to identify the domains of interaction. Nevertheless, we feel that this is an excellent future set of experiments that we will explore in greater detail and thank the reviewer for this suggestion. Moreover, we have amended the Discussion to include these experiments as a potential future direction. Please see subsection “Regulation of VHL in cancer”, end of first paragraph.

**Author response image 3. respfig3:** Co-IP of Daam2 truncation mutants (see Lee, et al. 2012) and VHL. These data show that each of the Daam2 truncation mutants is capable of associating with Daam2.

Previous studies from the Deneen laboratory suggest that DAAM2 regulates Wnt signaling in oligodendrocytes. Was this pathway excluded in the context of tumorigenesis?

This a very important point made by the reviewer, as it’s the driving force behind the RPPA experiments that led to the connection between Daam2 and VHL. This point was addressed in the original submission – please see Figure 4—figure supplement 1. In this figure we show that overexpression of Daam2 +Wnt3a ligand (similar conditions were used in Lee, et al. Developmental Cell 2012) did not have a profound effect on Wnt-reporter activity in glioma cell lines. Moreover, we introduced the Wnt-reporter into our mouse model and found that GOF and LOF of manipulations of Daam2 did not affect the number of cells demonstrating high levels Wnt activity. Together, these data indicate that in the context of human and mouse glioma, Daam2 does not robustly influence canonical Wnt-signaling.

Reviewer #2:[…] 1) This is not the first study to show an upstream mechanism of VHL regulation in cancer. The authors correctly point out in their Discussion that ID2 has been shown to regulate VHL, there is also a miRNA story in the literature (Li, et al. 2017). Thus while these other observations do not take away from the story here, this point is overstated several times in the text and should be corrected. Likewise, a comment in the Discussion implies that absent of other alterations, it is unclear how VHL function is inactivated in cancer, and the simple answer is tumor hypoxia.

This is an excellent point made by the reviewer and we thank her/him for raising this issue and pointing us to the miRNA paper (Li, et al. 2017). We have made the suggested changes in both the Introduction and Discussion. Please see Introduction, second paragraph and subsection “Regulation of VHL in cancer”, first paragraph for the amended text.

2) Figure 1; high resolution pictures of Daam2 staining that are in the supplemental data should be included in the figure.

We have made the suggested changes and have included the high-resolution images from the supplemental data as suggested. For the sake of simplicity and figure aesthetics, we have swapped the high-resolution images for the lower resolution images from the tissue array. The tissue array images are now in Figure 1—figure supplement 1.

3) Better descriptions of the two GBM cell lines would be useful. Do these contain similar driver mutations as the mouse models? Also, quantification of the colony assays would be helpful, rather than simply a representative figure. Do the control cells form colonies at all? Given that they form tumors, one would expect so.

This is also an important point and we thank the reviewer for bringing this issue to light. We have quantified the relative number of colonies that form in these assays, please see Figure 2. The control cells also form colonies. Figure 2 were taken at the point that Daam2-GOF group has shown colonies while the colonies in the control group is not visible yet.

4) Quantification of IHC for example for CD31 would be more convincing than a single picture.

We agree with the reviewer that quantification is critical for interpreting these results. We have included quantification of all CD31 staining using a recently established imaging method described in Tian, et al. Nature 2017. Please see Figure 4—figure supplement 3, K for these data.

5) HIF elevation in Figure 4 begs the question as to whether HIF target genes are also elevated in cells with altered Daam2.6) Can the authors speculate as to how Daam2 is functioning to facilitate VHL ubiquitylation? Is it an E3 ligase? Is there a domain structure of Daam2 that would support a hypothesis?

We thank the reviewer for bringing this issue to light, as further validation of dysregulated HIF1a target genes can only reinforce our model of Daam2 suppression of VHL function in glioma. Towards this, we performed Western blots on protein lysates extracted from CRISPR-IUE tumors that overexpress Daam2 (and appropriate controls), finding that these tumors exhibit drastic increases in VEGFA and Glut1 protein expression (both are bona fide direct targets of HIF1a). These data can be found in Figure 4 (see subsection “Daam2 promotes tumorigenesis through VHL”, first paragraph)and lend additional support to our model that Daam2 suppresses VHL function.

Reviewer #3:[…] A broad body of labor-intensive and technically challenging work is summarized here. The manuscript is well written. The conclusions would, in principle, be of interest to a broad readership in neuro-oncology and in other fields such as tumor metabolism and molecular therapeutics. Unfortunately, some of the data in this manuscript are, at present, inadequate to support the central claims. My concerns are as follows:1) Figure 2. The claim of this figure is that Daam2 overexpression accelerates the growth of human glioma cell lines. However, the authors do not actually show that Daam2 is overexpressed. They should provide a western blot towards this end. It is actually a bit curious that infection with lentivirus Daam2 has such a pronounced effect since the authors show us (in Figure 1) that gliomas already have high Daam2.

This is an excellent point raised by the reviewer and an obvious oversight on our part. We have included a representative Western blot demonstrating overexpression of ectopic Flag-tagged Daam2 in the glioma cell lines. Please see Figure 2.

Regarding native levels of Daam2 in these cell lines, as shown in Figure 1 Daam2 demonstrates heterogeneous expression across both human GBM and LGG. We confirmed this in silico observation using tissue microarrays, finding that a significant proportion of GBM and LGG tumors (~25%) demonstrate very low levels of Daam2 expression (Figure 1). Next we assessed Daam2 expression via qRT-PCR across a cohort of primary GBM cell lines, finding that amongst the 6 cell lines at our disposal, 2 demonstrated very low expression, 2 had intermediate expression, and 2 had very high expression (Author response image 4). Thus, these cell lines also demonstrate heterogeneous expression of Daam2, likely reflecting the nature of its expression in primary tumors from which they were derived. Based on these observations, we chose to perform our studies on GBM lines 1-2 because they have relatively low levels of Daam2 expression, compared to the rest of the cell lines, and therefore gave us the best opportunity to measure the effects of elevated Daam2 in this context.

**Author response image 4. respfig4:** qRT-PCR showing relative expression of Daam2 across 6 primary GBM cell lines.

2) Figure 3. An important claim of this figure is that Daam2 loss-of-function impairs growth of human glioma cells. The loss-of-function was achieved via shRNAi-mediated knockdown but there are two problems with the data. First, there is no western blot to show that Daam2 actually was knocked down. Second, there are growing concerns about off-target effects of shRNAi. The authors need a "rescue" experiment with Daam2 construct that is immune to their shRNAi to rule out off-target toxicity.

This is also an excellent point raised by the reviewer. We are unable to use Western blotting to measure reductions in Daam2 expression in these cell lines because the antibodies at our disposal are not effective at detecting human or mouse Daam2. Thus, as an alternative measure of Daam2 knockdown, we used qRT-PCR and these data are shown in Figure 3.

This is an excellent suggestion, as in this day and age scrambled controls are simply not enough of a control for shRNAi knockdown experiments. We performed rescue experiments using overexpression of mouse Daam2 in the shRNAi-knockdown cell lines. Fortunately for us, the shRNAi sequences that target human Daam2 are sufficiently divergent from the mouse cDNA and we are able to generate robust overexpression of mouse Daam2 in these lines. As shown in Figure 3—figure supplement 1, overexpression of mouse Daam2 is sufficient to rescue the in vitro proliferation defects caused by shRNAi knockdown of Daam2 in both GBM cell lines. These data indicate that the in vitro growth defects are the result of reduced expression of Daam2 caused by the shRNAi knockdown. We hope that this experiment satisfies the reviewer’s request for rescue of the Daam2-shRNAi associated phenotypes in these cell lines.

3) Figure 4—figure supplement 3. The central claim of this figure is that angiogenesis, as marked by CD31, is altered in response to Daam2 levels. The immunostain images shown are not convincing. The authors need to quantify these data.

This is a critical point that was also raised by reviewer 2. We have included quantification of all CD31 staining using an established imaging method described in Tian, et al. Nature 2017. Please see Figure 4—figure supplement 3, K for these data.

4) Figure 5. The central claim of these two figures is that ectopic VHL can overcome the accelerated growth induced by overexpression of Daam2 alone. As per comment #1 above, we need western blots – in this case for both VHL and Daam2. It would be interesting to see if the relationship between Daam2 and VHL is symmetric. If high Daam2 suppresses VHL, does high VHL suppress Daam2?

We thank the reviewer for this suggestion, as Western blots are a key component of these experiments. Towards, this we have included Western blots showing ectopic overexpression of Daam2 and VHL in both the human glioma cell lines and the mouse glioma models as well. Please see Figure 5 and Figure 5.

The issue of whether VHL can in turn suppress/degrade Daam2 is a potentially interesting avenue to pursue and we thank the reviewer for making this suggestion. Given that VHL functions as an E3 ligase, it is indeed possible that it reciprocally degrades Daam2. To investigate this we transfected cells with very high levels of Daam2/VHL, added cyclohexamide, and used Western blots to assess the levels of Daam2 expression under these conditions. As shown in Author response image 5, while the levels of VHL protein decreased over the 6-hour interval, the levels of Daam2 did not (Flag-Daam2). This suggests that either VHL is not capable of degrading Daam2 or that Daam2 has an extremely long half-life. While it is possible that VHL degrades Daam2, we were unable to detect this phenomenon. Nevertheless, we have addressed this issue in the Discussion, please see subsection “Regulation of VHL in cancer”, second paragraph.

**Author response image 5. respfig5:** High levels of VHL do not reciprocally degrade Daam2.

5) Figure 6. The central claim of this figure is that Daam2 forms a complex with VHL thereby promoting ubiquitination and degradation of VHL. These are very important mechanistic claims. I have some doubts about these data. These are:a) The antibody pulldown experiment needs to be shored up to rule out an adventitious interaction between Daam2 and VHL. Can the pulldown be reversed? Will IP VHL bring down Daam2? Is there any independent evidence of a VHL:Daam2 interaction, e.g. yeast two hybrid trap or proximity ligation assay?

This is a critical point and we thank the reviewer for raising this issue. We have performed the reverse IP and now show that VHL can bring down Daam2, please see Figure 6.

Regarding other evidence of Daam2/VHL interaction, we have not performed proximity ligation or a two-hybrid trap. While we agree that additional and independent assays measuring interaction would be ideal, we did not perform these experiments. With that said, we are very careful in the manuscript not to imply a direct interaction between Daam2/VHL because we never prove this. Instead throughout the manuscript we use “associate” to describe the biochemical relationship between Daam2/VHL. We feel that this description best represents our ability to detect co-IP of Daam2/VHL in both directions.

b) Figure 6. The central claim here is that VHL degradation is "significantly enhanced" in the presence of Daam2. I disagree with the descriptor of "significant". The offset in VHL T_1/2_ (5.08hr vs 3.68hr) might be "statistically significant" as the authors state but the overall difference is nuanced and does not seem adequate to account for the striking differential in VHL immunostain as shown in Figure 4. Author's comments?

The reviewer makes a good point about the disconnect between the in vivo observations and the in vitro degradation assays. First, while the offset in VHL half-life may seem nuanced, it is nonetheless statistically significant across multiple replicated experiments. Moreover, in the associated text we describe VHL degradation as “*enhanced*” in the presence of Daam2, which we feel is an apt description of the phenomenon, please see subsection “Daam2 facilitates ubiquitination of VHL protein”, last paragraph. With that said, one potential explanation for the disconnect between the in vivo and in vitro observations are the differences in systems. The in vitro assay was performed in 293 cells, while the in vivo observations were made in glioma models. It is possible that 293 cells are missing critical co-factors that participate in this process.

Another consideration is time. In the in vivo systems, Daam2 manipulation occurs across the lifetime of the resultant tumor (approx. 3 weeks for the PB-Ras model and 9-12 weeks for CRISPR-model). In contrast the in vitro experiments, occur across 6 hours. Given these differences in experimental duration, it is possible that we were able to witness more pronounced changes in VHL expression in vivo because these changes had more time to accumulate. In essence what we are witnessing in vivo (and in human tumors as well) is an accumulation of Daam2-mediated degradation over the life of the tumor.

Overshadowing all of these concerns with the data is a more fundamental question about the role of Daam2 as a non-genetic driver of gliomagenesis. What is the mechanism of leads to high expression of Daam2 in gliomas (Figure 1). Is Daam2 expressed at hyperphysiologic levels in these tumors or, alternatively, do some gliomas simply arise from Daam2-positive cells? If the latter is true than Daam2 is only permissive for gliomagenesis (along with many other things). If the former is true, then we need to know the mechanism.

This is indeed an important issue that we should address. Our interpretation of this question is the following: Are increases in Daam2 expression in glioma the result of i) active dysregulation or ii) a passive by-product of its expression in a cell lineage that is over-represented in glioma?

Regarding the possibility that Daam2 is actively dysregulated, it’s important to point out that we have detected increases in its mRNA levels (see Figure 1).

This raises the question of how Daam2 mRNA is elevated in these tumors and there are three likely possibilities: 1) direct transcriptional regulation, 2) microRNA regulation, or 3) regulatory links to key glioma drivers.

1) Regarding direct transcriptional regulation, we have no insight into how Daam2 is transcriptionally regulated. We are currently performing an enhancer screen in the chick spinal cord (similar to Kang, et al. Neuron 2012 and Glasgow, et al. Nature Neuroscience 2017) to identify regulatory elements that can be used as entry points for identifying transcriptional mechanisms that regulate Daam2 during development. These studies are in their nascent stages and it will take a while to have a definitive answer for the developmental mechanism; even longer to apply this mechanism to glioma.

2) Regarding microRNAs, there is evidence that mir335 can regulate Daam1/2 in neuroblastoma cell lines (Lynch, et al. PLoS One 2013), but whether this mechanism is applicable to glioma remains undefined and is beyond the scope of this study.

3) Another possibility is that Daam2 mRNA expression is linked in some way to key drivers of glioma that are frequently mutated (i.e. NF1, PTEN, EGFR, PDGFR, p53, Rb, etc.). Leveraging existing databases, we have correlated mutations in these key drivers with levels of Daam2 expression and there are no obvious or strong correlations.

These potential mechanisms are likely contributing, in some way, to Daam2 dysregulation in glioma, however as we have illustrated, deciphering these mechanisms will require a significant amount of work that is beyond the scope of this manuscript. With that said, we have amended the Discussion to include these areas as key future avenues of investigation. Please see subsection “Daam2 stimulates glioma tumorigenesis”, last paragraph.

The other possibility we must consider is that elevated Daam2 expression is the result of its expression in cell lineages overrepresented in glioma. Daam2 is expressed in both glial lineages: oligodendrocytes and astrocytes, in both precursor and mature stages of development. Both of these lineages are present in one form or another within the bulk tumor, therefore it is possible that elevated Daam2 expression is the result of analogous cell populations that comprise the primary, bulk tumor. Recently we found that glioma is comprised of non-stem cell, heterogeneous subpopulations of cells that are molecularly analogous to normal astrocyte subpopulations (Lin, et al. Nature Neuroscience 2017). Using the RNA-Seq data generated from these astrocyte subpopulations and their malignant counterparts, we found that Daam2 is uniformly expressed in all astrocyte and glioma subpopulations. These data suggest that, as far as we can tell, Daam2 is not selectively expressed in subpopulations of astrocytes or glioma cells that are overrepresented in glioma.

Another glioma subpopulation worthy of discussion is the glioma stem cell (GSC). We have not assessed Daam2 expression in GSC populations, as the markers that define this population in vivo can be nebulous. Moreover, the glioma stem cell comprises a very small fraction of the bulk tumor (~5-15%) therefore even if Daam2 was highly expressed in this population it is unlikely to account for the increased expression of Daam2 in glioma.

Despite these limitations, understanding the cellular basis for Daam2 function in glioma is also an important consideration. We have amended the Discussion to include these points as future directions,Please see subsection “Daam2 stimulates glioma tumorigenesis”, last paragraph.